# Nonlocal field theory of quasiparticle scattering in dipolar Bose-Einstein condensates

Caio C. Holanda Ribeiro[1] and Uwe R. Fischer[1]

[1]*Seoul National University, Department of Physics and Astronomy,*
*Center for Theoretical Physics, Seoul 08826, Korea*
(Dated: August 22, 2022)

We consider the propagation of quasiparticle excitations in a dipolar Bose-Einstein condensate, and derive a nonlocal field theory of quasiparticle scattering at a stepwise inhomogeneity of the sound speed, obtained by tuning the contact coupling part of the interaction on one side of the barrier. To solve this problem *ab initio*, i.e., without prior assumptions on the form of the solutions, we reformulate the dipolar Bogoliubov-de Gennes equation as a singular integral equation. The latter is of a *novel hypersingular* type, in having a kernel which is hypersingular at only two isolated points. Deriving its solution, we show that the integral equation reveals a continuum of evanescent channels at the sound barrier which is absent for a purely contact-interaction condensate. We furthermore demonstrate that by performing a discrete approximation for the kernel, one achieves an excellent solution accuracy for already a moderate number of discretization steps. Finally, we show that the non-monotonic nature of the system dispersion, corresponding to the emergence of a roton minimum in the excitation spectrum, results in peculiar features of the transmission and reflection at the sound barrier which are nonexistent for contact interactions.

## I. INTRODUCTION

How do the excitations of a quantum field above a given vacuum, the *quasiparticles*, propagate in a system which has nonlocal and anisotropic particle interactions? This seemingly simple question can be connected in fact to a plethora of related issues in different branches of physics and mathematics because of the very nature of the particle interactions. Indeed, in early quantum field theory, nonlocal alternatives for field theories were sought by the scientific community within the program of "handling divergences" [1, 2], and foundations of such nonlocal quantum field theories were extensively studied, with particular emphasis on S-matrix properties cf., e.g., Refs. [3–5]. However, the success and simplicity in studying low-energy phenomena afforded by the renormalization program of local quantum field theories eventually won as the primary paradigm. Nevertheless, instances of such nonlocal field theories appear, e.g., as necessary tools in investigating electromagnetic phenomena in material media [6–9], for studying trans-Planckian physics [10] and the universality of Hawking radiation [11], and to assess the influence of boundaries [12–14].

To study such nonlocal field theories in the lab, dipolar Bose-Einstein condensates (BECs) [15], realized in the quantum optical context of ultracold gases, cf., e.g., [16–18], offer a rich environment [19]. For example, quantum fluctuations in dipolar condensates, which lead to a peculiar Lee-Huang-Yang equation of state [20, 21], and the associated behavior of the thermodynamic pressure can lead to droplet stabilization, as observed in [22] and supersolid behavior [23], see for a review [24]. The droplet stabilization is becoming particularly intricate in the case of quasi-one-dimensional (quasi-1D) dipolar gases cf., e.g., Refs. [25–27].

Among the signatures of the dipolar interaction of particular importance for our analysis is the existence of rotonic excitations [28], which are caused by the anisotropy of the interaction, which is partly positive and partly negative. Roton modes, in particular, occur when the dipolar interaction dominates the interaction at high enough densities of the atoms or molecules. They play a pivotal role in the description of the dynamical instability emerging in the dimensional crossover from dynamically stable quasi-1D [29] or quasi-2D [30] condensates to 3D dipole-dominated BECs, which are always dynamically unstable. We focus in what follows on quasi-1D trapping.

The appearance of a roton minimum in the dimensional crossover signals a marked departure of the standard Bogoliubov dispersion relation from its contact interaction form, which is what is obtained in a local field theory. Together with the associated maxon maximum, it corresponds to a *non-monotonic* dispersion. Indeed, examples of intriguing effects related to rotonic excitations include the enhancement of many-body entanglement [31], of density oscillations [32], and the occurrence of roton confinement [33]. On the experimental side, rotons in elongated BECs have been observed, e.g., in [34–36].

We are however not aware of a solution of the Bogoliubov de Gennes (BdG) equation describing quasiparticle propagation in dipolar Bose-Einstein condensates in an inhomogeneous setup, e.g., presenting an interface between regions of distinct quasiparticle spectra. Finding such solutions is key to provide a general answer on the apparently basic question posed at the beginning of this Introduction, and we provide in the below an *ab initio* answer, in which we put the inhomogeneous dipolar BdG equation of a quasi-1D gas into the form of an equivalent singular integral equation, and solve this equation. To the best of our knowledge, this singular integral equation is *novel*, in that it provides an extension of the well known Cauchy-type singular kernels [37]. Specifically, the integral kernel we obtain is a combination of Cauchy-type kernels almost everywhere, with the exception of two *isolated* points where the singularity is stronger and

the kernel becomes hypersingular. Our case is however different from established textbook examples of hypersingular kernels [38], where the set of singular points has nonzero measure. From a more practical perspective, no universally reliable numerical method exists for solving singular integral equations, and each case must be treated differently in order to avoid numerical instabilities [37]. We discuss a method suitable for the BdG equation in its hypersingular integral form. We also provide a discretized version of the singular integral equation for the inhomogeneous dipolar BdG equation, and demonstrate its excellent performance for already a moderate number of discretization steps.

To give some intuition why the solution of this problem is nontrivial, note that within the instantaneous approximation for the dipolar interactions, a signal sent towards the barrier will interact with it before and after the signal has reached it. This is in striking contrast to the standard contact interaction case, where the signal interacts with the barrier only locally. Therefore, we expect nontrivial scattering phenomena to emerge. As we will show, these nontrivial phenomena are even more pronounced when roton excitations are involved due to the then increased number of the types of elementary excitations present in the system. Indeed, to assume an inhomogeneous configuration (e.g., a gas containing a sound barrier), as we will show, greatly increases the mathematical complexity of the perturbations in such systems, which in general forbids a fully analytical treatment.

Below, we reveal in detail how quasiparticles propagate in systems containing a sound barrier. Our results represent a major step for constructing a complete nonlocal field theory of dipolar BECs. The particular model we study, which encapsulates all required features, is a trapped BEC at rest with aligned magnetic or electric dipoles, which provides a sound barrier constructed by tuning locally the contact interaction between its particles, which is then separating the system into two regions with distinct sound velocities. Our goal is to make solutions to the nonlocal dipolar BdG equations as analytically amenable as possible. We show how the solutions we find can be used to build the S-matrix, which in the context of wave scattering comprises the reflection and transmission coefficients in such a way that unitarity is manifest. We shall see that when the dipolar interactions are present and the roton minimum exists, the increased number of the types of elementary excitations present in the system implies that the dimension of this matrix is larger than the $2 \times 2$ S-matrix for the case of contact-only interactions.

We shall discuss two methods of solving the BdG equation, one based on an approximate model, and the second one given in terms of special functions solutions to the novel class of singular integral equations we put forth. The method based on approximating the model has the advantage of allowing for analytic solutions, while the singular integral equation is treated numerically. Our results show that whenever the barrier exists, the dipolar interactions give rise to a continuum of evanescent channels bound to the barrier, which potentially play a role in near-boundary physics like recently explored in the context of the analogue gravity of sonic black holes [39]. Furthermore, novel characteristic features include a decrease in the barrier's transmittance when the roton minimum is about to form and the barrier's complete transmittance/reflectance for particular signals when the roton minimum exists even in the limit of "weak" barriers. These findings represent a remarkable departure from the homogeneous system (no barrier), where one may naively expect to see a continuous dependence of sound propagation on barrier height. Yet, complete transmittance/reflectance is observed even for vanishing barriers, near the roton and maxon frequencies, in marked disagreement with the continuous dependence obtained in contact-interaction condensates.

In summary, our presentation is organized in three parts as follows.

— The complexity of the scattering problem for our sound barrier model can be traced back to the analytical properties of the dipolar interactions in Fourier space. In section II we present the interaction kernel in Fourier space for a quasi-1D dipolar condensate and show that this kernel is *always* nonanalytical.

— In section III we present our condensate configuration containing a sound barrier for its sound waves, and solve the scattering problem. Specifically, in subsection III A we write down the BdG equation which models the propagation of small disturbances over our condensate. In subsection III B we classify all the quasiparticle excitations of our condensate, from which the scattering problem is formulated in terms of waves sent towards the barrier and the associated reflected and transmitted parts. Moreover, subsections III C and III D contain the methods we employ to solve the scattering problem.

— Lastly, we discuss in sections IV and V the physical implications of the quasiparticles found in section III. We show that the scattering process is unitary, and determine in detail how sound waves are scattered by interfaces in dipolar condensate.

Previous studies have dealt with phonon scattering and the associated S-matrix for contact interactions, e.g. in the context of acoustic Hawking radiation [40]. Yet, to the best of our knowledge we present the first complete *ab initio* nonlocal field theory of quasiparticle scattering at an inhomogeneity in the presence of a Bose-Einstein condensate on top of which the quasiparticles reside. We reveal, in particular, the impact of the anisotropy of interactions and the existence of a roton minimum on the scattering matrix. While a recent study explored the scattering properties of quasiparticles in polar dielectrics [41], our results are more general, do not assume any a priori

knowledge of boundary conditions imposed by the dipolar metamaterial geometry and constitution and incorporate, in distinction to [41], the existence of a condensate. Considering a dipolar BEC with a stepwise discontinuous contact interaction, the structure of the S-matrix is derived from first principles. Therefore, our study has, as a further application, the potential to describe metamaterials built from dipolar BECs, by establishing a clear recipe of how to predict scattering phenomena in such systems. It thus paves the way towards a plethora of applications obtained by generalizations of our model. For instance, our results can be readily applied for an inhomogeneous extension of the recent experiment reported in [42]. In this work, the crossover regime of a dipolar condensate to a supersolid and isolated droplet regimes was obtained by tuning the contact interaction, and studied using Bragg scattering of high energy excitations. By tuning the contact interaction locally, our model predicts the system response at any energy scale.

## II. DIPOLAR INTERACTIONS IN QUASI-1D CONDENSATES

### A. Interaction kernel after dimensional reduction

As explained in the introduction, dipolar condensates can be stabilized by trapping potentials, which avoid the head-to-tail instability of bulk dipolar condensates by dimensional reduction. In this section we discuss the trapping mechanism we adopt, for which our condensate behaves as a quasi-1D stable condensate. We start with a strongly elongated dipolar condensate with its dipoles oriented along a given direction $\boldsymbol{d}$ ($|\boldsymbol{d}| = 1$), such that its particles interact via the long-range instantaneous interaction energy

$$H_\mathrm{d} = \frac{C_\mathrm{dd}}{8\pi} \int \mathrm{d}^3x \mathrm{d}^3x' |\Phi(t, \boldsymbol{x})|^2 U_\mathrm{d}(\boldsymbol{x} - \boldsymbol{x}')|\Phi(t, \boldsymbol{x}')|^2, \quad (1)$$

in terms of the order parameter $\Phi$ and dipolar interaction strength $C_\mathrm{dd}$ [43]. The interaction kernel $U_\mathrm{d}$ is given by

$$U_\mathrm{d}(\boldsymbol{x}) = \frac{\boldsymbol{x}^2 - 3(\boldsymbol{x} \cdot \boldsymbol{d})^2}{|\boldsymbol{x}|^5}. \quad (2)$$

Let us assume the system is subjected to a strong radially symmetric trapping potential in such a way that the order parameter separation ansatz $\Phi(t, \boldsymbol{x}) = \phi_\perp(|\boldsymbol{x}_\perp|)\phi(t, x)$ holds, where $\phi_\perp$ is normalized as $\int \mathrm{d}^2x_\perp |\phi_\perp|^2 = 1$, assuming the geometry presented in Fig. 1. For the particular case of dipolar interactions, *under the assumed radially symmetric trapping*, it was shown [44] that the only contribution from the interaction kernel in Eq. (1) is given by the Fourier transform

$$U_\mathrm{d}(\boldsymbol{x}) = \frac{4\pi}{3(2\pi)^3} \left(1 - \frac{3}{2}\sin^2\theta\right) \int \mathrm{d}^3k \mathrm{e}^{i\boldsymbol{k}\cdot\boldsymbol{x}} \left(\frac{3k_x^2}{\boldsymbol{k}^2} - 1\right), \quad (3)$$

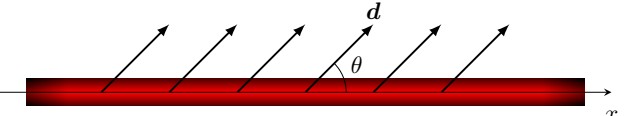

FIG. 1. Schematics of the elongated dipolar condensate under consideration. The system symmetry axis is here taken to be the $x$ axis, and the dipoles are oriented by an external field along the direction $\boldsymbol{d}$, which defines the angle $\theta$ as shown.

where $\theta$ is the angle between $\boldsymbol{d}$ and the $x$ axis, see for an illustration Fig. 1. It then follows from the order parameter separation ansatz that ($\Delta x = x - x'$)

$$H_\mathrm{d} = \frac{g_\mathrm{d}}{2} \left[ \int \mathrm{d}x |\phi|^4 - 3 \int \mathrm{d}x \mathrm{d}x' |\phi(x)|^2 G(\Delta x)|\phi(x')|^2 \right], \quad (4)$$

where $G$ is defined via its Fourier transform $\tilde{G}$ as [45]

$$\tilde{G}(\ell_\perp k_x) = \frac{\ell_\perp^2 k_x^2}{2} \int_{-\infty}^\infty \mathrm{d}k_\perp \frac{\mathrm{sgn}(k_\perp)\mathfrak{X}(\ell_\perp^2 k_\perp^2)}{k_\perp + ik_x},$$

$$\text{where} \quad \mathfrak{X}(\ell_\perp^2 \boldsymbol{k}_\perp^2) \coloneqq \frac{|\tilde{n}_\perp(\boldsymbol{k}_\perp)|^2}{2\pi\ell_\perp^2 \int \mathrm{d}^2x_\perp |\phi_\perp|^4}, \quad (5)$$

and with $\tilde{n}_\perp(\boldsymbol{k}_\perp) = \int \mathrm{d}^2x_\perp \exp(-i\boldsymbol{k}_\perp \cdot \boldsymbol{x}_\perp)|\phi_\perp(|\boldsymbol{x}_\perp|)|^2$. Here, $\ell_\perp$ denotes the typical length scale of the transverse trapping. Moreover, we have set as the effective quasi-1D dipole coupling

$$g_\mathrm{d} = g_\mathrm{d}(\theta, \ell_\perp) = -\frac{C_\mathrm{dd}}{3} \left(1 - \frac{3}{2}\sin^2\theta\right) \int \mathrm{d}^2x_\perp |\phi_\perp|^4. \quad (6)$$

We note at this point that $g_\mathrm{d} > 0$ is required for the system to be stable in the thermodynamic limit and for vanishing contact interaction.

We also observe that the advantages of presenting the dipolar kernel in Fourier space [Eq. (5)] rather than in configuration space are two-fold. Indeed, in Fourier space the double integral appearing in the energy functional (4) becomes a single integral (by the convolution theorem) which tends to simplify the analysis in particular when the condensate density is constant. Moreover, as we aim to study wave scattering at sound barriers, it is necessary to work with the interaction kernel in Fourier space, i.e., expressed in such a way that the role played by wave vectors become manifest.

For the (commonly employed) particular case of a strong harmonic trapping, one has the Gaussian approximation $|\phi_\perp(|\boldsymbol{x}_\perp|)|^2 = \exp(-\boldsymbol{x}_\perp^2/\ell_\perp^2)/(\pi\ell_\perp^2)$, where then $\ell_\perp$ is the harmonic oscillator length). For harmonic trapping, $\tilde{G}(\eta) = (\eta^2/2)\exp(\eta^2/2)E_1(\eta^2/2)$, $E_1$ being the first exponential integral function [44]. In our work one particular property of this function plays an important role: it has a discontinuity branch on the imaginary axis of the complex $k$ plane, which (for the Gaussian trans-

verse profile) comes from the function $E_1$ [46]. This feature increases the mathematical complexity of the condensate perturbations when some form of sound barrier exists in comparison to the case of contact-only interactions, and before we proceed to the model, let us pinpoint the origin of such a discontinuity and how it is related to the reduction to the quasi-1D regime. It is, in particular, not a feature of the transverse harmonic trapping per se, but occurs generically for any radial trapping, e.g., also for cylindrical box traps.

### B. Analyticity of the kernel

Assuming an analytical interaction kernel (in Fourier space) is a simplifying hypothesis in nonlocal field theories [11]. Indeed, for if $\tilde{G}(\ell_\perp k)$ is an analytical function of $k$, it follows that

$$\int \mathrm{d}x' G(\Delta x) f(x') = \tilde{G}(-i\ell_\perp \partial_x) f(x), \qquad (7)$$

for any function $f$, which leads us to question whether this property is fulfilled by our $\tilde{G}$. We note, in particular, that if Eq. (7) holds, the effect of the long-range interactions on plane waves is "local," and therefore the latter does not prevent the existence of plane wave solutions to the field equations. For the particular case of a dipolar interaction, inspection of Eq. (5) reveals the analytical structure of $\tilde{G}$ in the complex plane. It has a discontinuity branch on the imaginary axis as can be seen from the application of the Sokhotski-Plemelj identity $1/(q \pm i\epsilon) = 1/q \mp i\pi\delta(q)$ [47] as $k_x$ approaches the imaginary axis. Indeed, if $iq$ for real $q$ is any point on the imaginary axis, then straightforward manipulations lead to the jump magnitude measured by $\Delta\tilde{G}(iq) := \lim_{\epsilon\to 0}[\tilde{G}(iq + \epsilon) - \tilde{G}(iq - \epsilon)]$, which reads

$$\Delta\tilde{G}(iq) = i\pi q^2 \mathrm{sgn}(q)\mathfrak{X}(q^2). \qquad (8)$$

The above equation highlights the advantage of writing the interaction kernel in the integral form of Eq. (5), as it shows that the branch of $\tilde{G}$ exists for any shape of radial trapping, which in turn determines the discontinuity branch jump through the form factor $\mathfrak{X}$ in Eq. (8). For the Gaussian profile, the latter reads $\mathfrak{X}(q^2) = \exp(-q^2/2)$, while for a cylindrical box trap, for which $|\phi_\perp(|\boldsymbol{x}_\perp|)|^2 = 1/(\pi\ell_\perp^2)$ for $|\boldsymbol{x}_\perp| < \ell_\perp$ and zero otherwise, we find $\mathfrak{X}(q^2) = 2J_1^2(|q|)/|q|^2$, where $J_1$ is a Bessel function [46].

By tracing back from Eq. (5) to Eq. (3), we see that the existence of this discontinuity branch comes from the poles of the dipolar interaction in Fourier space, and Eq. (8) reflects the fact that different radial wavevectors add up to form the quasi-1D system. This is manifest in Eq. (3), where the pole at $\boldsymbol{k}^2 = k_x^2 + \boldsymbol{k}_\perp^2 = 0$ (in Fourier space) is evident. This gives rise to two first order poles at $k_x = \pm i k_\perp$ (manifest in Eq. (5)), which upon integration produces the discontinuity branch. In conclusion,

both the dipolar interaction and the dimensional reduction combine to give rise to the discontinuity branch.

The relevance of Eq. (8) to our model is that it greatly modifies the structure and complexity of the perturbations in the system when a sound barrier exists, mainly because the commonly assumed Eq. (7) *does not* hold. Moreover, while noting that we are ultimately interested in the case where the kernel describes dipolar interactions, these same conclusions also hold for a gas of charged bosons (e.g., a Cooper pair gas in a superconductor) whose pairwise interaction follows the Coulomb law. The latter, however, which represents isotropic interactions, does not give rise to a roton minimum. Finally, the particular details of the trapping mechanism enter the analysis only through $|\phi_\perp(|\boldsymbol{x}_\perp|)|^2$, which we assume henceforth to be given by the Gaussian approximation. We shall also omit the subscript $x$ from the momentum $k_x$ along the weakly confining direction and denote it as $k$ in what follows.

## III. QUASIPARTICLES IN THE PRESENCE OF A SOUND BARRIER

### A. Formulation of the Bogoliubov de Gennes problem

We consider a stationary background condensate at rest given by $\phi = \sqrt{n}\exp(-i\mu t)$ ($\hbar = 1$), with particle density $n$ and chemical potential $\mu$. In order to model a sound barrier for the phonons in this system, we also allow the particles to interact via the Feshbach-tunable contact term $H_c = g_c \int \mathrm{d}x |\phi|^4/2$, for an almost everywhere constant $g_c$ with a steplike discontinuity at $x = 0$. Then, as the local sound velocity is defined as $c = \sqrt{n(g_c + g_d)}$ (setting the mass of the dipolar atoms or molecules $m = 1$), we see that this setup corresponds to a system in which the sound velocity has a sudden jump — the sound barrier — at $x = 0$. Alternatively, from the sound velocity definition, we see that a barrier is also created for an inhomogeneous condensate with a particle density $n$ jump at $x = 0$, for fixed $g_c$. Bearing in mind that this simplified physical system already requires a complex mathematical treatment, we shall assume at once that the region for $x < 0$ is dipole-dipole dominated $g_c/g_d \sim 0$ and for $x > 0$, we have $g_c/g_d > 0$. Moreover, we shall assume for the sake of simplicity that the region $x > 0$ is such that $g_c$ prevents the formation a rotonic excitations. The discussion that follows can be easily extended to include the case in which rotonic excitations exist on both sides of the barrier, and also to sound barriers created by a density jump.

Therefore, the dynamics of the condensate is ruled by the system total energy $H := H_0 + H_c + H_d$, where $H_d$ is defined in Eq. (4),

$$H_0 = \int \mathrm{d}x \phi^* \left(-\frac{\partial_x^2}{2} + U\right)\phi, \qquad (9)$$

and $U$ is the 1D external potential, that leads to the 1D nonlocal Gross-Pitaevskii equation

$$i\partial_t\phi = \left[-\frac{\partial_x^2}{2} + U + (g_c + g_d)|\phi|^2\right]\phi - 3g_d\phi(G*|\phi|^2),$$
(10)

where $(G*|\phi|^2)(x) = \int dx' G(\Delta x)|\phi(x')|^2$ denotes the convolution. In deducing Eq. (10), we used that $G(\Delta x) = G(-\Delta x)$, as follows from the 3D dipolar kernel (2). Substituting our ansatz $\phi = \sqrt{n}\exp(-i\mu t)$ in Eq. (10) fixes the external potential in terms of the particle density:

$$U = \mu + \frac{\partial_x^2\sqrt{n}}{2\sqrt{n}} - n(g_c + g_d) + 3g_d G*n.$$
(11)

We note that the term $(\partial_x^2\sqrt{n})/2\sqrt{n}$ vanishes for the case of a homogeneous condensate, and equals a "delta derivative" potential for the case of a sudden varying $n$ at $x = 0$. The latter was recently explored by [39] in the context of analogue gravity in BECs.

Small disturbances in a stationary condensate are modeled by the Bogoliubov expansion $\phi = \exp(-i\mu t)(\sqrt{n} + \psi)$, where $|\psi|^2 \ll n$, and $\psi$ is a solution of the Bogoliubov-de Gennes (BdG) equation, obtained by linearizing Eq. (10):

$$i\partial_t\psi = \left(-\frac{\partial_x^2}{2} + \frac{\partial_x^2\sqrt{n}}{2\sqrt{n}}\right)\psi + n(g_c + g_d)(\psi + \psi^*)$$
$$- 3\sqrt{n}g_d G*[\sqrt{n}(\psi + \psi^*)].$$
(12)

In what follows, we take $n$ to be constant, in which case the BdG equation simplifies to

$$i\partial_t\psi = -\frac{\partial_x^2}{2}\psi + n(g_c + g_d)(\psi + \psi^*) - 3ng_d G*(\psi + \psi^*).$$
(13)

We emphasize, however, that the results below can be straightforwardly extended to the more general case of both $n$ and $g_c$ changing at the barrier.

We scale from now on lengths with $\xi_d = \sqrt{1/ng_d}$, wavevectors with $1/\xi_d$, and frequencies with $1/\xi_d^2$, assuming thereby that $g_d$ is always rendered finite and positive. When thus fixing the scale $\xi_d$, it should be kept in mind that $g_d$ depends on both the dipole orientation angle $\theta$ and the transverse trapping scale $\ell_\perp$ via Eq. (6).

Our goal in this work is to study the solutions of Eq. (13). They are more easily found in terms of the Nambu field $\Psi = (\psi, \psi^*)^t$, as demonstrated in detail for our type of system in [39]. Note that because of stationarity, the quasiparticle modes still assume the general form $\Psi(t,x) = \exp(-i\omega t)\Psi_\omega(x)$, and $\Psi_\omega$ satisfies

$$\omega\sigma_3\Psi_\omega = \left[-\frac{\partial_x^2}{2} + \left(1 + \frac{g_c}{g_d}\right)\sigma_4\right]\Psi_\omega - 3\sigma_4 G*\Psi_\omega,$$
(14)

where $\sigma_i$, $i = 1, 2, 3$ are the usual Pauli matrices, with $\sigma_4 = \mathbb{1} + \sigma_1$. As usual, if $\Psi_\omega$ is a solution for Eq. (14),

then $\sigma_1\Psi_\omega^*$ is also a solution with $-\omega^*$. Accordingly, we might focus on the field modes with $\omega > 0$. However, the analytical properties of $\tilde{G}$ prevent $\Psi_\omega$ from being a finite combination of exponential functions when $g_c$ is discontinuous (see for details Appendix A).

Far from the interface the solutions simplify to (a combination) of plane waves of the form $\Phi_k\exp(ikx)$ for constant $\Phi_k$, where

$$\left\{\omega\sigma_3 - \frac{k^2}{2} - \left[1 + \frac{g_c}{g_d} - 3\tilde{G}(\beta k)\right]\sigma_4\right\}\Phi_k = 0.$$
(15)

Here, the dimensionless parameter $\beta = \ell_\perp/\xi_d = \sqrt{\ell_\perp^2 ng_d(\theta, \ell_\perp)}$ (reinstating here $\xi_d$ for clarity) measures the extent to which we are in the quasi-1D regime (in the dipole-dominated case). The proper quasi-1D limit, with all perpendicular motion frozen out, is achieved when $\beta \to 0$.

The corresponding Bogoliubov dispersion relation is conveniently written as $f_\omega(k) = 0$, where

$$f_\omega(k) = \omega^2 - k^2\left[1 + \frac{g_c}{g_d} - 3\tilde{G}(\beta k) + \frac{k^2}{4}\right] = 0,$$
(16)

as we show in Appendices A and B.

We shall present below two routes for obtaining the solutions of Eq. (14). The solution for this equation is demonstrated in subsection III C, and a model approximation in which we discretize the integral of Eq. (5) is put forth in subsection III D.

## B. Classification of quasiparticles

We can enumerate all the possible solutions of Eq. (14) as presented in Fig. 2 in which each plane wave propagating towards the barrier corresponds to a distinct quasiparticle.

From Fig. 2, we see the characteristic roton minimum formation in the dispersion relation caused by the dipole interactions [28–30, 44, 48], which singles out the spectrum subset defined by $\Omega^{(r)} < \omega < \Omega^{(m)}$. For the sake of organization, we shall denote by $k$ (respectively $p$) the wavevector solutions at $x < 0$ (respectively $x > 0$). Note that, in contrast to the contact-only interaction case, in each region the rotonic dispersion relation always admits 6 possible wavevector solutions, and for the propagating ones, each corresponding group velocity sign (graph slope) indicates if the solution represents plane waves traveling towards or away from the interface at $x = 0$. Accordingly, if $\omega \notin (\Omega^{(r)}, \Omega^{(m)})$, we have only one solution at $x \ll 0$ (respectively $x \gg 0$) propagating towards the interface, denoted by $k_{in}$ (respectively $p_{in}$), and one solution propagating away from it, $k_1$ (respectively $p_1$). Moreover, we also find at each side of the boundary evanescent channels, i.e., channels that are exponentially suppressed far from the barrier, denoted by $k_2$, $k_3$, $p_2$, and $p_3$. These channels have $\text{Im } k_i < 0$ and $\text{Im } p_i > 0$, $i = 2, 3$.

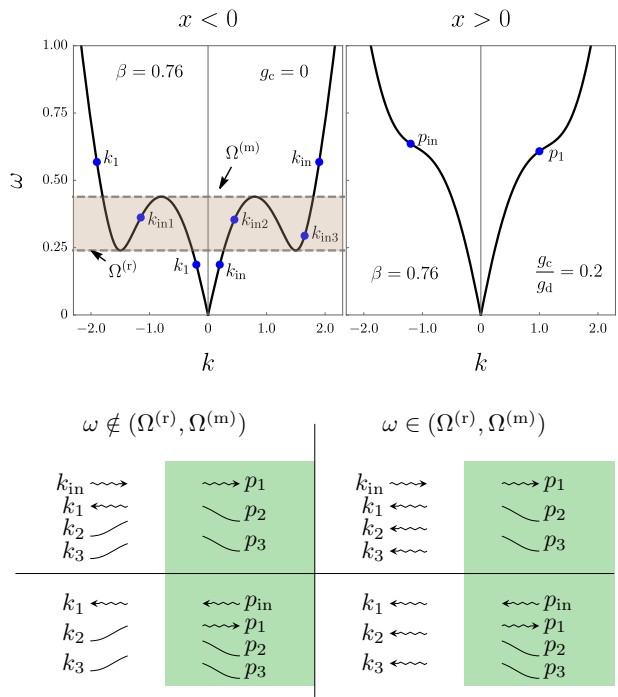

FIG. 2. Upper panel: Bogoliubov dispersion relation. Left: solutions in the region $x \ll 0$, characterized by a fully dipole dominated interaction; $\Omega^{(r)}$ and $\Omega^{(m)}$ are the roton and maxon frequencies, respectively. Right: solutions for the region $x \gg 0$, where a contribution from contact interaction is present. Lower panel: Asymptotics of all possible quasiparticle modes for the system under study. Zigzagged (respectively curved) lines represent propagating (respectively evanescent) channels. We note that $k_{\rm in}$ in the right panel represents the three possible choices $k_{\rm in1}$, $k_{\rm in2}$, and $k_{\rm in3}$. Also, arrows pointing to the right (respectively left) have $V_g > 0$ (respectively $V_g < 0$), where $V_{\rm g} = {\rm d}\omega/{\rm d}k$. Lower panel top part (respectively bottom part): schematics of the modes initiating at $x < 0$ (respectively $x > 0$). The scattering problem corresponds to a determination of how the transmission and reflection occurs for each possible incoming channel $k_{\rm in}$ and $p_{\rm in}$.

The complementary case — $\omega \in (\Omega^{(r)}, \Omega^{(m)})$ — is the physically richer one. We notice from Fig. 2 that all the six solutions at the left hand side of the barrier represent propagating waves, with three of them, $k_{\rm in1} < k_{\rm in2} < k_{\rm in3}$ propagating towards the barrier. We shall label the channels propagating away from the barrier as $k_1 < k_2 < k_3$. To summarize the above discussion, we depict in Fig. 2 lower panel the schematics of all solutions of Eq, (14), from which we can state the main result of our work — the solution of the scattering problem —, as follows: For each wave sent towards the barrier at $x = 0$, we show how to determine the intensity of the reflected and transmitted waves.

The distinct behavior for dipolar interactions when a roton minimum is present comes from the shaded region on the LHS of the top panel Fig. 2, where three modes $k_{\rm in1}$, $k_{\rm in2}$, and $k_{\rm in3}$ in the band between $\Omega^{(r)}$ and $\Omega^{(m)}$, the

roton and maxon frequencies, respectively, can propagate towards the barrier (cf. lower panel on the left). This is in marked distinction to the contact-dominated case on the right ($x > 0$) of the barrier.

We note that the existence of complex wave vector solutions to the dispersion relation is not a peculiarity of our dipolar condensate as they are also present in inhomogeneous contact-dominated configurations [39]. Furthermore, evanescent channels admit no physical interpretation in term of quasiparticles, i.e., they, alone, do not represent solutions to the BdG equation *per se*. The same, however, is not necessarily true for real wave vector solutions, for at least in configurations similar to the one we considered in which an asymptotic regime exists (far from the barrier), propagating channels can be directly linked to the quasiparticles of the asymptotic system. The interpretation of each component appearing in a quasiparticle mode is thus model dependent. In the system we considered the novelty regarding evanescent channels is their increased number, which from a physical perspective is an indication that near the barrier the transient regime caused by the barrier existence is distinct from the contact condensate analogue, as expected from the non-local character of the interactions.

## C. Singular integral equation for the dipolar Bogoliubov de Gennes problem

The details of how to solve Eq. (14) are presented thoroughly in Appendix B, and here we synthesize the main points. Following the asymptotic behavior just presented, we know that the quasiparticle modes are labeled by each incoming signal ($k_{\rm in}$ or $p_{\rm in}$), and far from the barrier they reduce to linear combinations of the channels shown in Fig. 2 lower panel. Our strategy here can be understood in terms of the latter channels as follows. For local, contact-only condensates, quasiparticle modes are built by combining the plane waves given by the Bogoliubov dispersion relation, i.e., local solutions to the BdG equation, and imposing matching conditions at the barrier. This procedure fails in the dipolar case, as the plane waves from the dispersion relation are not local solutions to the BdG equation. Accordingly, we shall develop a procedure to build, from the plane waves of Fig. 2 local solutions which can then be combined in the same fashion as for contact-interaction condensates. This "completion" procedure, performed via the BdG equation, gives rise to singular integral equations and associated special functions, but has the benefit of expressing the quasiparticle modes in a manner that leaves manifest their asymptotic properties while allowing for an easier numerical treatment than the direct solution of the BdG equation.

Because each field mode has a continuum of evanescent channels as imposed by the convolution in Eq. (14) (Appendix A), solutions can be found with the aid of the

ansatz

$$\Psi_\omega = \sum_k S_k \zeta_k(x)\Phi_k + \sum_p S_p \zeta_p(x)\Phi_p, \qquad (17)$$

where the $k's, p's, \phi_k$, and $\phi_p$ are given by Eqs. (15) and (16). The quantities $\zeta_k$ and $\zeta_p$ are matrix-valued func-

tions, given by

$$\zeta_k(x) = \begin{cases} i\int_0^\infty \mathrm{d}q \Lambda_{k,q} e^{-qx}\Pi(q), \ x > 0, \\ e^{ikx} - i\int_{-\infty}^0 \mathrm{d}q \Lambda_{k,q} e^{-qx}\Pi(q), \ x < 0, \end{cases} \quad (18)$$

and

$$\zeta_p(x) = \begin{cases} -e^{ipx} + i\int_0^\infty \mathrm{d}q \Lambda_{p,q} e^{-qx}\Pi(q), \ x > 0, \\ -i\int_{-\infty}^0 \mathrm{d}q \Lambda_{k,q} e^{-qx}\Pi(q), \ x < 0. \end{cases} \quad (19)$$

with $\Pi(q) = \left(q^2/2 - \omega\sigma_3\right)\sigma_4/(q - q_-)(q - q_+)$. Furthermore, the functions $\Lambda_{k,q}$ and $\Lambda_{p,q}$ are solutions of the novel singular integral equation

$$\frac{h(q)\Lambda_{k,q}}{(q - q_-)(q - q_+)} + \frac{3i\Delta\tilde{G}(i\beta q)}{2\pi(q_- - q_+)}\left[\frac{1}{q - q_-}\int_{-\infty}^\infty \mathrm{d}q' q'^2 \Lambda_{k,q'}\left(\frac{1}{q_- - q'} - \frac{1}{q - q'}\right) - \{q_- \leftrightarrow q_+\}\right] = -\frac{3i\Delta\tilde{G}(i\beta q)}{2\pi(iq - k)}, \quad (20)$$

where the integrals are Cauchy principal values, $\Delta\tilde{G}(iq)$ was defined in Eq. (8), and the function $h(q)$ is defined by

$$h(q) = \frac{q^4}{4} - \omega^2 - q^2\left[1 + \frac{g_c(q)}{g_d} - 3\overline{G}(i\beta q)\right], \quad (21)$$

with $\overline{G}(iq) = \lim_{\epsilon \to 0^+}[\tilde{G}(iq + \epsilon) + \tilde{G}(iq - \epsilon)]/2$, i.e., the average of $\tilde{G}$ along the discontinuity branch. Finally, the real parameters $q_- < 0 < q_+$ are the two simple zeros of $h$: $h(q_\pm) = 0$.

We stress that the ansatz (17) was constructed in such a way that each $\zeta_k(x)\Phi_k, \zeta_p(x)\Phi_p$ is a local solution to the BdG equation, i.e., they are built to satisfy Eq. (14) at all points except at the barrier ($x = 0$). Thus, the continuum of evanescent channels in Eqs. (18) and (19) gives a succinct representation of the fact that the Bogoliubov channels of Fig. 2 fail from being local solutions. Moreover, a few features of the ansatz (17) are revealed by direct inspection of Eq. (20). In general, for analytic interaction kernels, one has $\Delta\tilde{G}(iq) := 0$, which implies $\Lambda_{k,q}, \Lambda_{p,q} := 0$ for all $k$'s and $p$'s, and the ansatz (17), through Eqs. (18) and (19), reduce to a finite combination of the exponentials given by the Bogoliubov dispersion relation (16). This is, in particular, the case for contact-type interactions. Furthermore, the decay of $\Lambda_{k,q}, \Lambda_{p,q}$ as functions of $q$ depends on the radial trap through $\Delta\tilde{G}(iq)$. For the Gaussian approximation adopted here, Eq. (20) shows that $\Lambda_{k,q}, \Lambda_{p,q}$ are exponentially suppressed for large $q$, whereas for the box trap profile, $\Lambda_{k,q}, \Lambda_{p,q}$ decay with a power law [cf. Eq. (8) and the discussion after it].

In order to find a solution in the form (17), the scattering coefficients $S_k$ and $S_p$ must be uniquely fixed up to an overall phase and a normalization constant. We note first that substitution of this ansatz into the BdG equation implies (after a lengthy calculation) that a solution

in this form exists only if

$$\sum_k S_k \Lambda_{k,q_-}\sigma_4\Phi_k + \sum_p S_p \Lambda_{p,q_-}\sigma_4\Phi_p = 0, \qquad (22)$$

$$\sum_k S_k \Lambda_{k,q_+}\sigma_4\Phi_k + \sum_p S_p \Lambda_{p,q_+}\sigma_4\Phi_p = 0, \qquad (23)$$

are satisfied, which thus fixes two of the six scattering coefficients. These equations are necessary conditions for Eq. (20) to hold (details in Appendix B). The remaining boundary conditions are fixed by standard wave mechanics techniques applied to Eq. (14): $\Psi_\omega$ and $\partial_x\Psi_\omega$ are continuous at the barrier.

A few more general comments about the ansatz (17) are in order. First of all, the solution just found is build from the solutions of the singular integral equation (20), which in fact can be even more difficult to solve than the BdG equation itself. However, this equation can be solved numerically to any precision by means of cubic splines [49]. Although this requires a considerable numerical effort, the latter method has an advantage over e.g. collocation schemes [50], as it does not rely on prior assumptions on the form of the solutions (for which explicit examples are contained in Appendix D). The sensitivity towards choosing the appropriate numerical scheme additionally serves to illustrate the mathematical challenge posed by nonlocal field theories.

Also, we see from the definitions in Eqs. (18) and (19) that in addition to the evanescent channels coming from the dispersion relation (16), the nonlocal dipolar interactions give rise to a continuum of evanescent channels *when a sound barrier exists*. This can be traced back to the analytical properties of $\tilde{G}$ discussed in Sec. II, as we see from there that if $\Delta\tilde{G} \neq 0$ (see Eq. (8)), then $\Lambda_{k,q}$ given by Eq. (20) is nonvanishing. Furthermore, we know that when the barrier is absent ($g_c = 0$), the

solutions to the BdG equation are single propagating exponentials [30, 31, 44]. The numerical implementation of the general solution (17) for this particular case recovers this fact, as we verified when building the numerical solutions presented in Appendix D.

We conclude this presentation of solutions to the BdG equation with a further remark regarding the level of mathematical complexity in this system when compared with sound propagation in contact-only interacting condensates. In the latter, the analogue situation of a sound barrier modeled by a sudden $g_c$ jump over a homogeneous condensate results in the quasiparticle modes being combinations of only a few simple exponentials [39]. This inspiration, gleaned from contact interactions, motivates us to seek for a different strategy in solving for the scattering coefficients, which consists in an approximation to the kernel (5), such that the corresponding quasiparticle modes are also expressed as a finite combination of exponentials.

### D. Approximate solution

The strategy we follow in this subsection consists in finding an approximate discrete version of the integral equation (20), in such a way that the exact quasiparticles can be recovered via some limiting process. It turns out that a convenient way of achieving such approximation is to substitute the dipolar kernel integral (5) by a finite sum. Specifically, we take $\tilde{G} \to \tilde{\mathcal{G}}$, where

$$\tilde{\mathcal{G}}(k) = \frac{k^2}{\sum_{j=0}^{\mathcal{N}} j\Delta q^2 e^{-j^2\Delta q^2/2}} \sum_{j=0}^{\mathcal{N}} \frac{j\Delta q^2 e^{-j^2\Delta q^2/2}}{j^2\Delta q^2 + k^2}, \quad (24)$$

and the two parameters $\Delta q > 0$ and the integer $\mathcal{N}$ are free. We note that $\mathcal{N} \to \infty$ and $\Delta q \to 0$ reproduces exactly Eq. (5), as depicted in Fig. 3. Furthermore, $2\mathcal{N}$ is the number of (simple) poles in this function, which are located at $k = j(i\Delta q)$, for $-\mathcal{N} \le j \le \mathcal{N}$, and, naturally, $\Delta q$ is the distance between two consecutive poles. The examples of Fig. 3 show the accuracy potential of our approximation. In particular, we see from Fig. 3 that for $\mathcal{N} = 10$ and $\Delta q = 1/3$, i.e., the approximating $\tilde{\mathcal{G}}$ containing only 20 poles, a reasonable agreement is already obtained. Let us now build the solutions for $\Psi_\omega$ for the model of Eq. (24).

The general asymptotic properties of the possible quasiparticles modes are the same as presented in Fig. 2. In the absence of $\tilde{\mathcal{G}}$, the dispersion relation in Eq. (16) is a degree four polynomial equation, whose solutions are explicitly found for each $\omega > 0$, whereas for $\tilde{\mathcal{G}}$ given in Eq. (24), in addition to these four solutions, another $2\mathcal{N}$ solutions are present, one for each pole in $\tilde{\mathcal{G}}$. Furthermore, in view of the numerator of Eq. (24), no dispersion relation solution coincides with the poles of $\tilde{\mathcal{G}}$ in the complex $k$ plane.

For the approximate model under study, the solutions for the dispersion relation can be grouped in the single

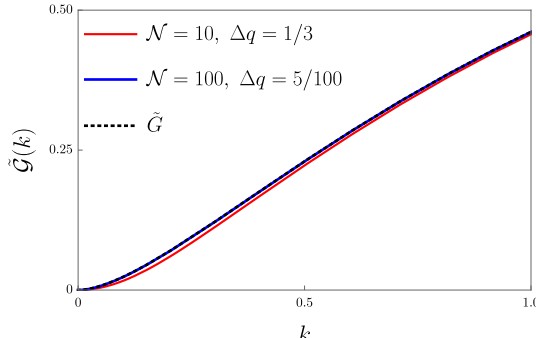

FIG. 3. Approximations for the Fourier space kernel $\tilde{G}$ in (5) for two sets of $\{\mathcal{N}, \Delta q\}$. For $\mathcal{N} = 100$ the approximation is indistinguishable on the scale of the figure from the exact result.

zero level set $f_\omega(k) = 0$, as done in Fig. 4. As explained, in each region of the condensate and for each $\omega > 0$, Eq. (16) has $4 + 2\mathcal{N}$ solutions, with the real solutions depicted in Fig. 2, and the remaining being necessarily complex. We recall that when $\tilde{G}$ is the exact one given in Eq. (5), the wave function presents a continuum of evanescent modes on the imaginary axis. We thus readily see that the approximation implemented here also discretizes this continuum, as indicated by Fig. 4, where in addition to the six channels existing when $\tilde{G}$ is used, we have the extra blue points on the imaginary axis. Furthermore, as $\mathcal{N} \to \infty$ and $\Delta q \to 0$, these poles clump together, thus recovering the continuum of the solutions (Eq. (17)) and reinforcing the validity of our approximation. We label the evanescent channels at $x < 0$ by $k_j$

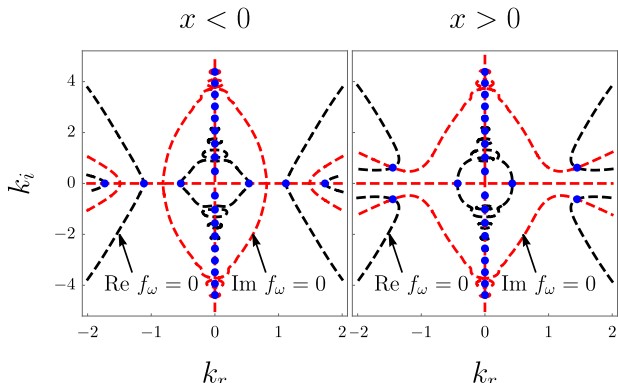

FIG. 4. Zero level sets for the real and imaginary parts of Eq. (16). Blue circles are solutions to $f_\omega = 0$ for the fixed frequency $\omega = 0.4$, i.e., they are the wave vectors solutions to the Bogoliubov dispersion relation in the complex plane. We have chosen $\mathcal{N} = 10$ and $\Delta q = 1/3$ in the discrete representation (24). $k_r$ and $k_i$ denote the real and imaginary parts of $k$, respectively. Left: (Right:) dispersion relation at $x < 0$ ($x > 0$). The solutions on the imaginary axis tend to a continuum of evanescent channels when $\mathcal{N} \to \infty$ and $\Delta q \to 0$.

and at $x > 0$ by $p_j$. Clearly, Im $k_j \leq 0$ and Im $p_j \geq 0$.

It turns out that these exponentials, the solutions to the dispersion relation Eq. (16) of the approximated model, can be used to construct solutions to the wave equation. Indeed, let us look for a solution with the ansatz

$$\Psi_\omega = \begin{cases} \sum_p S_p e^{ipx} \Phi_p, & x > 0, \\ \sum_k S_k e^{ikx} \Phi_k, & x < 0, \end{cases} \qquad (25)$$

whose relation to Eq. (17) is manifest: the integrals in Eqs. (18) and (19) are substituted by finite sums of evanescent channels (see Fig. 4). Each incoming signal at the barrier gives rise to a distinct solution, which is associated to reflected and evanescent channels. By carefully counting the number of unknown coefficients $S_k$ and $S_p$, we find that for a given $\mathcal{N}$, we have $2 + \mathcal{N}$ $S_k$'s and $2 + \mathcal{N}$ $S_p$'s, leading to a total $4 + 2\mathcal{N}$ unknown coefficients for each field mode. If a solution exists in the form (25), then the coefficients are fixed by the BdG equation (14). We note, however, that the application of standard wave mechanics techniques to the field equation only produces 4 boundary conditions, namely, $\Psi_\omega$ and $\partial_x \Psi_\omega$ are continuous at the barrier. This is caused because the convolution in Eq. (14) is always continuous (see Appendix A).

We conclude, therefore, that $2\mathcal{N}$ boundary conditions appear to be "missing," and the solution to this puzzle is found from the form of Eq. (25). Indeed, the convolution $\sigma_4 G * \Psi_\omega$ of Eq. (14) is conveniently written as

$$\frac{1}{2\pi} \int dk\, e^{ikx} \tilde{\mathcal{G}}(\beta k) \sigma_4 \tilde{\Psi}_\omega(k), \qquad (26)$$

where $\tilde{\Psi}_\omega(k)$ is the Fourier transform of $\Psi_\omega$ (cf. Appendix A), and the (simple) poles of $\tilde{\Psi}_\omega$ are precisely the $k$'s and $p$'s appearing in Eq. (25). Thus, as the poles of $\tilde{\mathcal{G}}(\beta k)$ cannot coincide with the poles of $\tilde{\Psi}_\omega$ (which are given by the dispersion relation (16)), the convolution (26) does not have evanescent channels at the poles of $\tilde{\mathcal{G}}(\beta k)$ if and only if $\sigma_4 \tilde{\Psi}_\omega$ vanishes at these poles. Therefore, as $\tilde{\mathcal{G}}(\beta k)$ has $2\mathcal{N}$ poles at the imaginary axis, a solution of the form (25) exists if and only if

$$\sigma_4 \tilde{\Psi}_\omega(ij\Delta q) = 0, \qquad (27)$$

for $-\mathcal{N} \leq j \leq \mathcal{N}$, $j \neq 0$. This gives us an additional set of $2\mathcal{N}$ boundary conditions, as required.

This concludes the construction of the quasiparticle modes for the system under study. We emphasize at this point that the approximate solutions described in the above greatly reduce the numerical effort necessary to simulate any observable quantity in the inhomogeneous dipolar BEC.

## IV. S-MATRIX AND UNITARITY

All the quasiparticle modes thus constructed admit the in/out state interpretation: For each field mode,

the incoming channel, labeled by $k_{\text{in}}$'s or $p_{\text{in}}$, represents a signal sent towards the boundary at the asymptotic past which emerges at the asymptotic future propagating away from the barrier through the various reflected/transmitted channels. This can be read off directly from the solutions in Eqs. (17) and (25), as far away from the boundary each elementary excitation reduces to a sum of plane waves. The scattering coefficients involved in all of these processes can be conveniently grouped into an unitary matrix — the S-matrix — which is simply an expression for the conservation of field mode normalization throughout the system's causal development. When only contact interactions are present, this conservation is implied by the BdG equation to be $\partial_x J_\omega = 0$, where $J_\omega = \text{Im} \, (\Psi_\omega^\dagger \partial_x \Psi_\omega)$. The important consequence of this equation is that the total flux in the system is conserved: $J_\omega(\infty) - J_\omega(-\infty) = 0$, giving rise to the S-matrix conveniently defined in terms of the norm-preserving condition

$$\sum_{k \text{ prop}} k \Phi_k^\dagger \Phi_k |S_k|^2 \stackrel{\text{contact}}{=} \sum_{p \text{ prop}} p \Phi_p^\dagger \Phi_p |S_p|^2, \qquad (28)$$

where the sums are performed over the propagating channels only. It is to be stressed that this conservation law is not due to the particle number conservation of the full theory implied by its $U(1)$ symmetry. It is a well known fact that the Bogoliubov expansion leads to a theory with a spontaneously broken $U(1)$ symmetry [51]. The conservation law we are exploring here is, then, the time-independence of the Bogoliubov scalar product, defined as $\langle \Psi, \Psi' \rangle := \int dx\, \Psi^\dagger \sigma_3 \Psi'$, where $\Psi = \exp(-i\omega t)\Psi_\omega(x)$, $\Psi' = \exp(-i\omega' t)\Psi_{\omega'}(x)$ are any two field modes. Via the BdG equation (14), the condition $\partial_t \langle \Psi, \Psi \rangle = 0$ can be shown to be equivalent to $\partial_x J_\omega = 0$ for all $\omega$ *in the absence of dipolar interactions.*

However, when dipolar interactions are present, $J_\omega$ is no longer conserved (see for the extended discussion in Appendix C):

$$\partial_x J_\omega = 6\text{Im}[(G * \Psi_\omega^\dagger)\sigma_4 \Psi_\omega]. \qquad (29)$$

This should not be read as implying that there is no flux conservation, but that the quantity $J_\omega$ is no longer a bona fide representation for the system total flux. We show in detail in Appendix C that for both families of solutions found in the previous section the dipolar analogue of Eq. (28) acquires the form

$$\sum_{k \text{ prop}} \Phi_k^\dagger \left( k - 3\frac{d\tilde{\mathcal{G}}}{dk}\sigma_4 \right) \Phi_k |S_k|^2 =$$

$$\sum_{p \text{ prop}} \Phi_p^\dagger \left( p - 3\frac{d\tilde{\mathcal{G}}}{dp}\sigma_4 \right) \Phi_p |S_p|^2. \qquad (30)$$

The meaning of this relation is more readily grasped by constructing the S-matrix. To that end, we take the so-

lution for $\Phi_k$ to be normalized as

$$
\Phi_k = \left| \frac{k^2}{4\pi V_{\mathrm{g}}\omega(\omega - k^2/2)^2} \right|^{1/2}
$$
$$
\times \left( \begin{array}{c} 1 + g_{\mathrm{c}}/g_{\mathrm{d}} - 3\tilde{\mathcal{G}} \\ \omega - k^2/2 - 1 - g_{\mathrm{c}}/g_{\mathrm{d}} + 3\tilde{\mathcal{G}} \end{array} \right), \qquad (31)
$$

where $V_{\mathrm{g}} = \mathrm{d}\omega/\mathrm{d}k$ is the group velocity.

We have presented here the flux conservation in terms of the approximated model, which in turn implies its validity for the continuum as we take the limit of an infinite number of discretization steps. Also, we note that the particular form of the normalization in Eq. (31) is not relevant for the physics of the problem. However, this form is convenient for us because it implies

$$
\Phi_k^\dagger \left( k - 3\frac{\mathrm{d}\tilde{\mathcal{G}}}{\mathrm{d}k}\sigma_4 \right) \Phi_k = \frac{1}{2\pi}\mathrm{sgn}(V_{\mathrm{g}}), \qquad (32)
$$

for propagating channels. This form of normalization is one possible choice which bounds the scattering coefficients absolute value to unity, as we shall see now. Let us label the quasiparticle modes according to their incoming channel by adding a superscript to the scattering coefficients. For instance, for $p_{\mathrm{in}}$, we set $S_k \to S_k^{p_{\mathrm{in}}}$ in the general solution (25). Moreover, we set to unity the intensity of the incoming channels, i.e., $S_{p_{\mathrm{in}}}^{p_{\mathrm{in}}} = S_{k_{\mathrm{in}}}^{k_{\mathrm{in}}} = 1$.

In view of the schematics displayed in Fig. 2 lower panel, Eqs. (30) and (32) simply mean that the matrix $\boldsymbol{S}_\omega$ defined by

$$
\boldsymbol{S}_\omega = \left( \begin{array}{cc} S_{k_1}^{k_{\mathrm{in}}} & S_{p_1}^{k_{\mathrm{in}}} \\ S_{k_1}^{p_{\mathrm{in}}} & S_{p_1}^{p_{\mathrm{in}}} \end{array} \right), \qquad (33)
$$

for $\omega \notin (\Omega^{(\mathrm{r})}, \Omega^{(\mathrm{m})})$ and

$$
\boldsymbol{S}_\omega = \left( \begin{array}{cccc} S_{k_1}^{k_{\mathrm{in}1}} & S_{k_2}^{k_{\mathrm{in}1}} & S_{k_3}^{k_{\mathrm{in}1}} & S_{p_1}^{k_{\mathrm{in}1}} \\ S_{k_1}^{k_{\mathrm{in}2}} & S_{k_2}^{k_{\mathrm{in}2}} & S_{k_3}^{k_{\mathrm{in}2}} & S_{p_1}^{k_{\mathrm{in}2}} \\ S_{k_1}^{k_{\mathrm{in}3}} & S_{k_2}^{k_{\mathrm{in}3}} & S_{k_3}^{k_{\mathrm{in}3}} & S_{p_1}^{k_{\mathrm{in}3}} \\ S_{k_1}^{p_{\mathrm{in}}} & S_{k_2}^{p_{\mathrm{in}}} & S_{k_3}^{p_{\mathrm{in}}} & S_{p_1}^{p_{\mathrm{in}}} \end{array} \right), \qquad (34)
$$

in case that $\omega \in (\Omega^{(\mathrm{r})}, \Omega^{(\mathrm{m})})$, is unitary, i.e., $\boldsymbol{S}_\omega^\dagger \boldsymbol{S}_\omega = \boldsymbol{S}_\omega \boldsymbol{S}_\omega^\dagger = \mathbb{1}$. This "unitarity" enables us to study quasiparticle scattering by the sound barrier as considered, because it ensures that no signal amplitude is lost during quasiparticle propagation and scattering. Finally, a different choice of normalization for $\Phi_k$ clearly does not spoil the S-matrix as it is given by Eq. (30), but the simplified form $\boldsymbol{S}_\omega$ and its unitarity condition changes.

We finish this section with a disclaimer regarding the S-matrix nomenclature. Naturally, scattering matrices appear generically in distinct branches of physics, and as such it is important that the meaning of the notion of S-matrix be completely clear in each context. We cite, for instance, the two works [52, 53] in which S-matrices appear within the same wave mechanics context as in our work. Yet, in [53] the very same mathematical object is

not referred to as an S-matrix. We should thus stress that the S-matrix in our context is not the same object as the one occurring in particle scattering in quantum field theory, where it is just representing the Schrödinger equation written in terms of the evolution operator [54]. Note that a proof of the unitarity of the S-matrix in such a quantum field theory context for a nonlocal family of scalar quantum field theories was provided in [5].

## V. TRANSMITTANCE AND REFLECTANCE: IMPACT OF DIPOLAR INTERACTION

As an application of the solutions constructed we now investigate how the sound barrier transmittance/reflectance is affected by the roton minimum. Once the scattering coefficients are known, we can study how waves sent towards the barrier get reflected and transmitted, the dipolar condensate analogue of the scattering of light rays at interfaces in classical optics. Specifically, far from the barrier ($|x| \gg 1$) in general we obtain from Eq. (17) or (25) that

$$
\Psi_\omega \sim \left\{ \begin{array}{ll} \sum_{p \text{ prop}} S_p e^{ipx}\Phi_p, & x > 0, \\ \sum_{k \text{ prop}} S_k e^{ikx}\Phi_k, & x < 0, \end{array} \right. \qquad (35)
$$

i.e., only the propagating channels (real wave vectors) contribute. For the sake of illustration, if we consider the case where no roton is present, the quasiparticle of frequency $\omega$ and indexed by $k_{\mathrm{in}}$ far from the barrier assumes the form

$$
\Psi^{k_{\mathrm{in}}} \sim e^{-i\omega t} \left\{ \begin{array}{l} S_{p_1}^{k_{\mathrm{in}}} e^{ip_1 x}\Phi_{p_1}, \; x > 0, \\ e^{ik_{\mathrm{in}}x}\Phi_{k_{\mathrm{in}}} + S_{k_1}^{k_{\mathrm{in}}} e^{ik_1 x}\Phi_{k_1}, \; x < 0. \end{array} \right. \qquad (36)
$$

This *is* the solution of the BdG equation in our dipolar condensate configuration. Indeed, for contact-only condensates the quasiparticles also assume the form in Eq. (36) far from inhomogeneities, and thus we conclude that the scattering coefficients encapsulate all new physical information coming from the dipolar interactions in comparison to contact interactions. We observe that Eq. (36) corresponds to a plane wave of "unity intensity" sent towards the barrier from the asymptotic left, namely the term proportional to $\exp(-i\omega t + ik_{\mathrm{in}}x)$, which then gives rise to a reflected wave $[\exp(-i\omega t + ik_1 x)]$ with intensity $|S_{k_1}^{k_{\mathrm{in}}}|^2$ and a transmitted wave $[\exp(-i\omega t + ip_1 x)]$ with intensity $|S_{p_1}^{k_{\mathrm{in}}}|^2$. Moreover, the unitarity of the process discussed in the previous section shows us that the constraint $|S_{k_1}^{k_{\mathrm{in}}}|^2 + |S_{p_1}^{k_{\mathrm{in}}}|^2 = 1$ holds. When the roton minimum is present, the same reasoning and interpretation holds, with the difference that more propagating channels might be present in the quasiparticles following the scheme of Fig. 2. We discuss in the below each of the possible cases separately.

We use in this section *only* the approximate model, as it allows for an easier numerical simulation, and leave for the Appendix D some worked out examples of the singular integral BdG equation (20). We recall from the

dispersion relation (16) that the influence of the dipolar interactions is measured by the coefficient $\beta = \ell_\perp$: As $\beta$ increases, the roton minimum emerges and the system eventually becomes dynamically unstable; when $\beta \to 0$, the system is stable under the proviso that $g_d > 0$. Therefore, $\beta$ measures how deep into the quasi-1D regime the system has penetrated.

Furthermore, we note from Eqs. (5) and (24) that, because of the global factor $k^2$, the long-range part of the dipolar interactions is suppressed in the quasi-1D limit $\beta \to 0$, and the condensate behaves no different than one with only local contact interactions, which is modeled by $g = g_d$ for $x < 0$ and $g = g_d + g_c$ for $x > 0$. Thus our model enables us to compare results with the non-dipolar case operating near or within the quasi-1D limit. For the sake of a clear representation, we now treat separately the cases with roton and no roton minimum.

### A. No roton minimum

When rotonic excitations are not present in the system, for each frequency $\omega$ in the system spectrum there are two elementary excitations, corresponding to the signals sent towards the barrier at each of its sides, i.e., signals $k_{\rm in}$ and $p_{\rm in}$. Accordingly, the S-matrix has the form (33) for each frequency subspace. As $\boldsymbol{S}_\omega$ is unitary, this means that both its row and column vectors form orthonormal bases, which in turn implies that the absolute value of one of its components fixes all the others. Therefore, by calculating $|S_{p_1}^{k_{\rm in}}|^2$ — the transmitted intensity through the barrier from left to right — we also determine $|S_{k_1}^{k_{\rm in}}|^2$, $|S_{p_1}^{p_{\rm in}}|^2$, and $|S_{k_1}^{p_{\rm in}}|^2$.

We show in Fig. 5 how the barrier transmittance varies with frequency for several values of $\beta$. The dotted-brown curve corresponds to $\beta = 0$, and it shows that the barrier with "height" modeled by $g_c/g_d = 0.2$ is almost transparent for contact-only interactions. Yet, we note that a noticeable decay in the transmittance is predicted to occur for an extended range of signal frequencies even in the absence of a roton minimum when $\beta$ is increased, which thus reveals how wave propagation in this system is sensitive to inhomogeneities.

### B. With roton minimum

When the roton minimum exists, however for frequencies not in the grey band shown in the left upper panel of Fig. 2, $\omega \notin (\Omega^{(r)}, \Omega^{(m)})$, there are only two quasiparticle excitations for each frequency, and the same reasoning used to interpret the case with no roton minimum present can be repeated. We present in Fig. 6 several simulations for this regime, which supplements our findings from Fig. (5) beyond the roton minimum formation, namely, the high energy sector of the theory is sensitive to the bending of the dispersion relation, a feature not

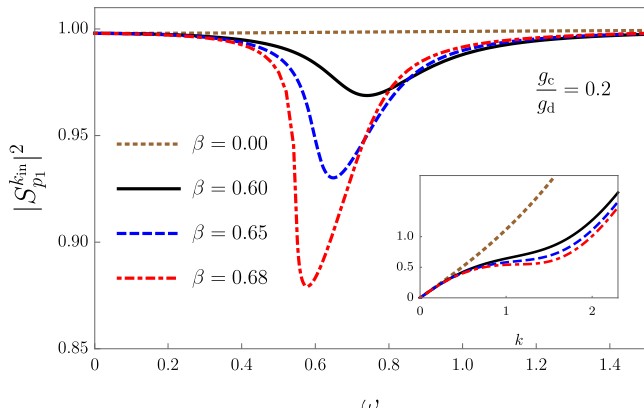

FIG. 5. Sound barrier transmittance measured by the coefficient $|S_{p_1}^{k_{\rm in}}|^2$ for several values of $\beta$. Inset: positive branch of the dispersion relation for each curve, showing how the roton minimum threshold is approached. The brown-dotted curve corresponds to the contact-only regime ($\beta = 0$), showing that this barrier is almost transparent for contact-only interactions. Yet, the bending of the dispersion relation leads to a noticeable decay of the barrier transparency even in the absence of a roton minimum, as shown by the continuous, dashed, and dot-dashed curves. Here, discretization parameters are $\mathcal{N} = 10$ and $\Delta q = 1/3.4$. For these parameters, the roton minimum formation threshold is approximately $\beta \sim 0.689$.

present when only contact interactions exist for condensates at rest.

Furthermore, within the band $\omega \in (\Omega^{(r)}, \Omega^{(m)})$ (the shaded area in Fig. 6) the degeneracy of each frequency subspace is 4 for the parameter choice we are investigating, corresponding to the four distinct quasiparticles that can be excited: $k_{\rm in1}$, $k_{\rm in2}$, $k_{\rm in3}$, $p_{\rm in}$. We stress that this increased degeneracy has no counterpart in condensates whose particles interact only locally. It is instructive to analyze each quasiparticle separately.

#### 1. $\omega \in (\Omega^{(r)}, \Omega^{(m)})$, $k_{\rm in1}$ excitation

We show in Fig. 7 the reflectance and transmittance coefficients of the quasiparticle branch indexed by $k_{\rm in1}$, cf. Fig. 2. Reflectance and transmittance measure the fractions of the plane wave signal $\exp(-i\omega t + ik_{\rm in1} x)$ coming towards the barrier from the left that get reflected and transmitted through the various available channels: $k_1$, $k_2$, $k_3$, and $p_1$. Therefore, the data in Fig. 7 show us that the barrier is mostly opaque for these waves, as the transmittance in this case satisfies $|S_{p_1}^{k_{\rm in1}}|^2 \ll 1$. Furthermore, the signal is integrally reflected through the channel $k_1$ (respectively $k_2$) for frequencies close to the roton (respectively maxon) frequency, and this feature survives even for very small $g_c/g_d = 0.001$. It is noteworthy that in the barrier's absence, i.e., $g_c = 0$, any signal of this form is clearly integrally transmitted, and *thus the very existence of the barrier leads to the total reflection of $k_{\rm in1}$*

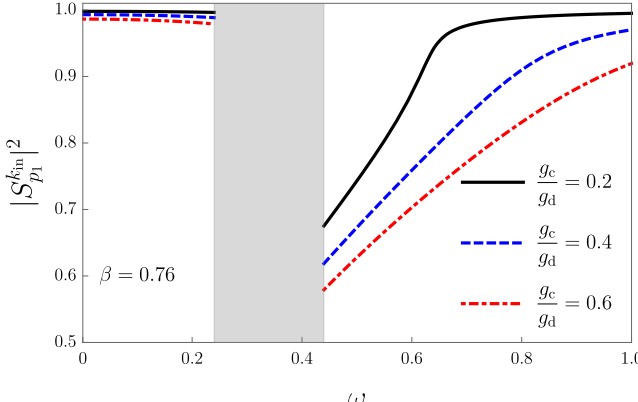

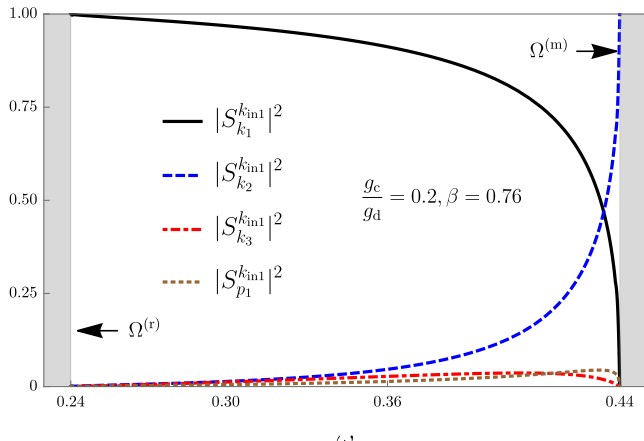

FIG. 6. Sound barrier transmittance measured by the coefficient $|S_{p_1}^{k_{in}}|^2$ for barrier heights $g_c/g_d$. The shaded region correspond to the band $\omega \in (\Omega^{(r)}, \Omega^{(m)})$, which cannot be characterized by the coefficient $|S_{p_1}^{k_{in}}|^2$ alone. Here, $\mathcal{N} = 10$, $\Delta q = 1/3.4$ and $\beta = 0.76$. This implies $\Omega^{(r)} \sim 0.24$ and $\Omega^{(m)} \sim 0.45$ for the roton and maxon frequencies, respectively. These curves represent supplementary data to the ones presented in Fig. 5: although those sound barriers have weak influence on the phonon sector of the theory, the high-energy sector is sensitive to the bending of the dispersion relation, a feature not present in the contact-only case.

FIG. 7. Scattering coefficients for the quasiparticle branch indexed by $k_{in1}$ in the frequency band $\omega \in (\Omega^{(r)}, \Omega^{(m)})$. Here, $\mathcal{N} = 10$, $\Delta q = 1/3.4$. This implies $\Omega^{(r)} \sim 0.24$ and $\Omega^{(m)} \sim 0.45$ for the roton and maxon frequencies, respectively. We note that the sum $|S_{k_1}^{k_{in1}}|^2 + |S_{k_2}^{k_{in1}}|^2 + |S_{k_3}^{k_{in1}}|^2 + |S_{p_1}^{k_{in1}}|^2 = 1$ for all $\omega$, one of the properties of the S-matrix. For the quasiparticles, labeled by $k_{in1}$, the barrier is mostly opaque, with the incoming signal being reflected exclusively through the channel $k_1$ for $\omega \to \Omega^{(r)}$ and through the channel $k_2$ as $\omega \to \Omega^{(m)}$.

*quasiparticles with frequencies near $\Omega^{(r)}$ and $\Omega^{(m)}$.*

### 2. $\omega \in (\Omega^{(r)}, \Omega^{(m)})$, $k_{in2}$ excitation

The analysis of the remaining quasiparticle modes follows the same line of reasoning just presented. Figure 8 presents the scattering coefficients for the $k_{in2}$ quasiparticle branch.

We see from Fig. 8 that, in distinction to the $k_{in1}$ quasiparticles, plane waves of the form $\exp(-i\omega t + ik_{in2}x)$ sent towards the barrier from the left are integrally transmitted for frequencies close to the roton minimum, and a transition to a completely opaque barrier is observed as the frequency tends to $\Omega^{(m)}$. We also observe that this opaqueness near maxon frequencies survives even for very small $g_c/g_d = 0.001$, in the same way as it happens for the $k_{in1}$ excitation discussed in the preceding paragraph.

### 3. $\omega \in (\Omega^{(r)}, \Omega^{(m)})$, $k_{in3}$ and $p_{in}$ excitations

The remaining quasiparticle branches, namely, $k_{in3}$ and $p_{in}$ are analyzed in the same fashion. They correspond to the plane waves $\exp(-i\omega t + ik_{in3}x)$, $\exp(-i\omega t + ip_{in}x)$ propagating toward the barrier from the left and right, respectively. We see from Fig. 9 that both waves experience a partially transmitting barrier for frequencies near $\Omega^{(m)}$, in distinction to the $k_{in1}$ and $k_{in2}$ excitations discussed above that experience a completely opaque barrier at these frequencies.

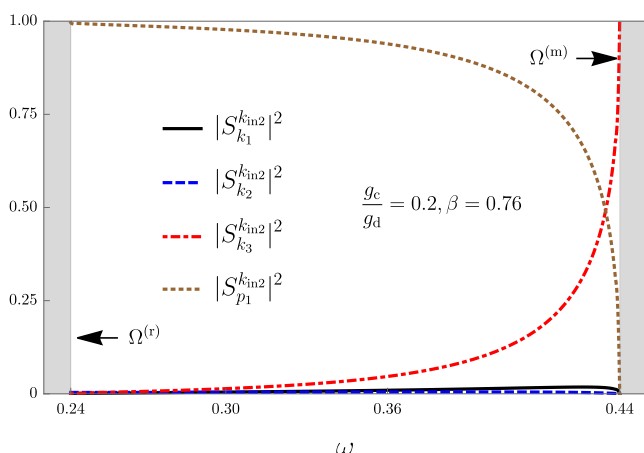

FIG. 8. Scattering coefficients for the quasiparticle branch indexed by $k_{in2}$ in the frequency band $\omega \in (\Omega^{(r)}, \Omega^{(m)})$. The parameters are the same as in Fig. 7. We observe that these quasiparticles experience a transition from a completely transparent barrier for frequencies near $\Omega^{(r)}$ to a completely opaque one, for frequencies near $\Omega^{(m)}$ and with the reflection exclusive through $k_3$. The channels $k_1$ and $k_2$ have negligible participation in the scattering process.

Furthermore, for frequencies near the roton minimum, the barrier becomes completely opaque for the $k_{in3}$ excitations and completely transparent for the $p_{in}$ branch.

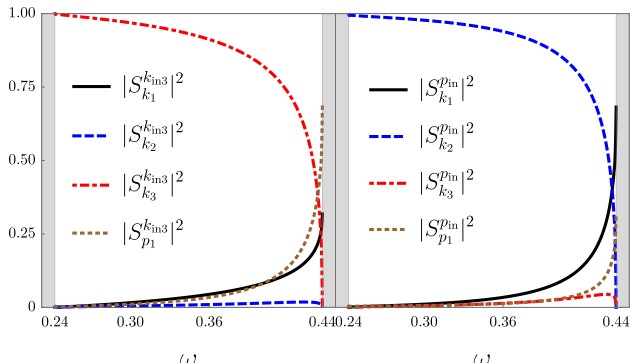

FIG. 9. Scattering coefficients in the frequency band $\omega \in (\Omega^{(r)}, \Omega^{(m)})$. Left panel: quasiparticle branch indexed by $k_{\text{in3}}$. Right: quasiparticle branch indexed by $p_{\text{in}}$. The parameters are the same as in Figs. 7 and 8. Both families of quasiparticles experience a partially transmitting barrier for frequencies near $\Omega^{(m)}$, whereas for frequencies near the roton minimum, the quasiparticles $k_{\text{in3}}$ (respectively $p_{\text{in}}$) experience a completely opaque (respectively transmitting) barrier.

## VI. FINAL REMARKS

The present study provides a systematic route to describe the scattering of quasiparticles in inhomogeneous dipolar Bose-Einstein condensates. To this end, we have studied perturbations in a quasi-1D dipolar condensate in which a sound interface exists, separating the condensate in two regions possessing different sound velocities. The perturbations were built via two distinct methods, one based on the direct simulation of the solutions and one based on a family of approximated models. Both methods explore the fact that the long-range dipolar interaction in Fourier space is not modeled by an analytic kernel, which gives rise to a distinct set of evanescent channels bound to the sound barrier.

As a particular application, we have shown how sound scattering occurs as a function of the perturbation frequency. An intricate pattern of reflectance/transmittance was shown to emerge due to the existence of rotonic excitations, which are due to the interaction anisotropy, which leads to a strong dependence of the barrier transmittance as function of the mode frequency and "polarization." In fact, if rotonic excitations exist, for frequencies within the roton/maxon band, the number of elementary excitations that can be scattered by the barrier is larger than two, as happens for the case where only contact interactions are present. Each of these correspond to a distinct system response to external excitations, and the barrier behaves fully/partially transparent or opaque depending on the excitations.

We started from a homogeneous quasi-1D dipolar condensate at rest, and asked how sound propagates in this system if a sound barrier is constructed at which the sound speed experiences a steplike change. As the system admits also rotonic (and maxonic) excitations when dipole-interaction-dominated, we demonstrated that the S-matrix dimension is enlarged. It should be emphasized that, starting from these two basic ingredients we use— dipolar condensate at rest and a sound barrier —, that only through the exact knowledge of the quasiparticle modes we derived it is possible to unveil the peculiar results presented for the reflectance and transmittance of the increased number of modes available to the system. The properties of the S-matrix we obtain clearly distinguish dipole-interaction-dominated BECs from their their contact-interaction-dominated counterparts.

The importance of the present work thus consists in providing a recipe to build the quasiparticle spectrum in dipolar condensates with inhomogeneous sound velocity. We should also stress that our method is not restricted to condensates with homogeneous densities, as only a few modifications are necessary to study configurations in which the sound interface is caused by density jumps. Indeed, once the knowledge of how to build quasiparticle mode solutions for this type of inhomogeneous dipolar condensates is set, it is straightforward to apply the same technique a number of systems of interest. In particular, the application we have explored here is far from exhausting all the features inherent in this type of system, with examples including a roton almost touching the wavevector axis which could hybridize with low-energy phonons, and the existence of rotonic excitations on both sides of the barrier. As a natural extension of our results, nontrivial effects are expected if the sound barrier is taken to have a finite size. Furthermore, we expect that selective nature of quasiparticle scattering in dipolar BECs through the associated transmission and reflection coefficients will have significant impact on the properties of droplets, which represent naturally emerging interfaces in the condensate due to quantum fluctuations [22, 24–27].

Our analysis indicates how intricate quasiparticle scattering in inhomogeneous dipolar BECs with continuous variations of density and coupling will be, where in general no simple solutions to the quasiparticle modes can be found. The extension of our technique to the case of continuous sound barriers therefore promises to reveal further features of quasiparticle scattering in dipolar BECs over those familiar from contact interaction condensates.

The quasiparticle modes we constructed are already properly normalized according to the Bogoliubov scalar product. Therefore, quantization of the present nonlocal field theory is a formal step which follows straightforwardly from expanding the bosonic field operator into this complete set of modes.

We finally note that infinitely extended (quasi-)1D Bose-Einstein condensates, according to the Hohenberg theorem [55], do not exist [56], due to the divergence of nonlocal phase fluctuations in the system. This imposes certain restrictions on the applicability of infinitely extended systems to model experimental realizations, as we expect the phonon part of the spectrum to be sensitive to finite size effects. This, however, should not in-

terfere with the general properties of the scattering processes here investigated — which occur far from the long-wavelength phonon regime — as long as the condensate is still sufficiently large along the weakly confining direction, where the achievable length is subject to a position space version of the Hohenberg theorem [57].

## VII. ACKNOWLEDGMENTS

We thank Dr. Luana Silva Ribeiro for her valuable comments regarding singular integral equations. This work has been supported by the National Research Foundation of Korea under Grants No. 2017R1A2A2A05001422 and No. 2020R1A2C2008103.

## Appendix A: A property of the convolution with $G$

In this appendix we show how $G * f$ differs from $\tilde{G}(-i\beta\partial_x)f$ when $f$ is given by

$$f(x) = \Theta(x)e^{ipx}, \tag{A1}$$

for any real number $p$, that is why the identity (7) does not hold in our system.

For analytic kernels, $G * f(x) - \tilde{G}(-i\beta\partial_x)f = 0$ always holds. Because we have a nonanalytic kernel for dipolar interactions, assessing the difference between $G * f$ and $\tilde{G}(-i\beta\partial_x)f$ is of importance in our analysis. Let $\tilde{f}(k) = \int dx \exp(-ikx)f(x) = i/(p-k+i\epsilon)$ be the Fourier transform of $f(x)$. Then from the convolution theorem we have $G * f(x) = (1/2\pi)\int dk \exp(ikx)\tilde{G}(\beta k)\tilde{f}(k)$, which can thus be straightforwardly evaluated by means of the residue theorem. The sole extra step comes, however, from the discontinuity branch of $\tilde{G}$ on the imaginary axis. We find

$$G * f(x) - \Theta(x)e^{ipx}\tilde{G}(\beta p) =$$
$$\frac{1}{2\pi}\begin{cases} \int_0^\infty dq e^{-qx}\Delta\tilde{G}(i\beta q)/(iq-p), & x > 0, \\ -\int_{-\infty}^0 dq e^{-qx}\Delta\tilde{G}(i\beta q)/(iq-p), & x < 0. \end{cases} \tag{A2}$$

We thus see that the RHS of Eq. (A2) differs from zero because $\tilde{G}$ is discontinuous on the imaginary axis. Equation (A2) also tells us that far from $x = 0$, where $f$ is discontinuous, $G * f(x) - \Theta(x)e^{ipx}\tilde{G}(\beta p) \sim 0$, as expected.

## Appendix B: Solution of Eq. (14)

In this appendix we present the ideas to obtain the general solution (17) of the main text. We recall that when $g_c$ is everywhere constant in Eq. (14), its solutions assume the form $\exp(ikx)\Phi_k$ for all $x$ and constant $\Phi_k$. However, when $g_c$ possesses the jump-like discontinuity we are considering in this work, the solutions obviously fail to assume these simple forms, and, in view of the discussion in Appendix A, only asymptotically far from the barrier the solutions are combinations of plane waves. Nevertheless, we know from Appendix A, Eq. (A2) that each plane wave excites, through the dipolar interactions, a continuum of evanescent modes bound to the barrier. This suggests we might look for a solution of the form

$$\Psi_\omega = \begin{cases} \sum_{k'} S_{k'}e^{ik'x}\Phi_{k'} - i\int_{-\infty}^0 dq e^{-qx}\Lambda_q, & x < 0, \\ -\sum_p S_p e^{ipx}\Phi_p + i\int_0^\infty dq e^{-qx}\Lambda_q, & x > 0, \end{cases} \tag{B1}$$

where the $k's$ and $p's$ label the asymptotic solutions, given by Eqs. (15) and (16), which are obtained by direct substitution into Eq. (14), with the aid of Eq. (A2). The function $\Lambda_q$ is a spinor, just like $\Psi_\omega$ itself.

After substitution of the above ansatz in Eq. (B1) into the Bogoliubov equation (14) and a lengthy calculation with repeated use of the residue theorem, we find that

$$\int_0^\infty dq e^{-qx}\left\{\left\{\frac{q^2}{2} + \omega\sigma_3 - \left[1 + \frac{g_c(q)}{g_d} - 3\bar{G}(i\beta q)\right]\sigma_4\right\}\Lambda_q + \frac{3i\Delta\tilde{G}(i\beta q)}{2\pi}\sigma_4\left(\int \frac{dq'\Lambda_{q'}}{q'-q} + \sum_{k'}\frac{S_{k'}\Phi_{k'}}{iq-k'} + \sum_p \frac{S_p\Phi_p}{iq-p}\right)\right\}$$
$$= 0, \tag{B2}$$

for all $x > 0$, and similarly for $x < 0$, with the $q$ integration running from $-\infty$ to 0. Also, the Cauchy principal value is to be taken for the $q'$ integral contained inside the round brackets. Eq. (B2) is true only if the term inside the outermost curly brackets vanishes, which then yields a Cauchy-type singular integral equation [50].

Equation (B2) can be further simplified by relating

$\Lambda_q$ to the scalar functions $\tilde{\Lambda}_{k',q}$ and $\tilde{\Lambda}_{p,q}$ occurring in Eqs. (18)-(20) of the main text,

$$\Lambda_q = \left(\frac{q^2}{2} - \omega\sigma_3\right)\sigma_4\left(\sum_{k'} S_{k'}\tilde{\Lambda}_{k',q}\Phi_{k'} + \{k' \leftrightarrow p\}\right), \tag{B3}$$

With this relation, we find that

$$0 = \sum_{k'} S_{k'} \sigma_4 \Phi_{k'} \left[ h(q)\tilde{\Lambda}_{k',q} + \frac{3i\Delta\tilde{G}(i\beta q)}{2\pi} \left( \int \frac{dq' q'^2 \tilde{\Lambda}_{k',q'}}{q' - q} + \frac{1}{iq - k'} \right) \right] + \{k' \leftrightarrow p\} \tag{B4}$$

where the function $h$ is defined in Eq. (21) of the main text.

A quick inspection of Eq. (B4) suggests that we could define each function $\tilde{\Lambda}$ to be the solution of the integral equation inside the square brackets, e.g.,

$$h(q)\tilde{\Lambda}_{k',q} + \frac{3i\Delta\tilde{G}(i\beta q)}{2\pi} \left( \int \frac{dq' q'^2 \tilde{\Lambda}_{k',q'}}{q' - q} + \frac{1}{iq - k'} \right) \stackrel{?}{=} 0, \tag{B5}$$

in such a way that the BdG equation would be satisfied at all points except at $x = 0$. However, this procedure is "partially" wrong, and the reason for this is that we must understand the space of solutions for these integral equations first. Note that if we remove the integral part in Eq. (B5), then the function $\tilde{\Lambda}_{k',q}$ necessarily has two poles, at $q = q_\pm$, defined by the two zeros of the function $h(q_\pm) = 0$. That happens because the operator which

multiplies by $h(q)$ is only invertible when restricted to the space of solutions that diverge at $q_\pm$. Because of this feature, we might look for solutions in the form

$$\tilde{\Lambda}_{k',q} = \frac{\Lambda_{k',q}}{(q - q_-)(q - q_+)}. \tag{B6}$$

By plugging this ansatz back into Eq. (B4), notice that the integral kernel becomes

$$\int \frac{dq' q'^2 \Lambda_{k',q'}}{(q' - q)(q' - q_-)(q' - q_+)}, \tag{B7}$$

which is not defined when $q \to q_\pm$ unless $\Lambda_{k',q_\pm} = 0$.

This kind of divergence is well understood and we can trace it back to the groundbreaking work of Ugo Fano [58]. Indeed, we have

$$\frac{1}{(q' - q)(q' - q_-)(q' - q_+)} = \frac{1}{q_- - q_+} \left[ \frac{1}{q - q_-} \left( \frac{1}{q_- - q'} - \frac{1}{q - q'} \right) - \{q_- \leftrightarrow q_+\} + \pi^2 \delta(q' - q)[\delta(q - q_-) - \delta(q - q_+)] \right], \tag{B8}$$

i.e., each double pole contains delta-contributions, lead-

ing to the divergence of the integral as $q \to q_\pm$. Now, in view of Eqs. (B6) and (B8), Eq. (B4) becomes

$$\sum_{k'} S_{k'} \sigma_4 \Phi_{k'} \left\{ \frac{h(q)\Lambda_{k',q}}{(q - q_-)(q - q_+)} + \frac{3i\Delta\tilde{G}(i\beta q)}{2\pi(q_- - q_+)} \left[ \frac{1}{q - q_-} \int dq' q'^2 \Lambda_{k',q'} \left( \frac{1}{q' - q} - \frac{1}{q' - q_-} \right) + \pi^2 q_-^2 \delta(q - q_-) \Lambda_{k',q_-} \right. \right.$$
$$\left. \left. - \{q_- \leftrightarrow q_+\} + \frac{q_- - q_+}{iq - k'} \right] \right\} + \{k' \leftrightarrow p\} = 0. \tag{B9}$$

We thus see that the term inside the first curly brackets in Eq. (B9), apart from the delta functions, defines the integral equation (20) of the main text. Referring back to Eq. (B2), we conclude that the ansatz (B1) is a solution of the BdG equation (14) only if the two delta functions contributions in Eq. (B9) at $q = q_\pm$ vanish. This leads precisely to the conditions Eqs. (22) and (23) of the main

text. We can then assert that the functions $\Lambda_{k',q}$ satisfy the integral equation (20), which can be solved numerically to any desired accuracy by means of cubic splines [49], see Appendix D.

## Appendix C: Flux conservation

In this appendix we shall present a proof for the flux conservation stated in Eq. (30). Due to the fact that $\tilde{\mathcal{G}}$ tends to $\tilde{G}$ in the limit of a large number of discretization steps, we demonstrate flux conservation for the discrete approximation, from which conservation for the continuum model then follows. We shall start by taking two arbitrary solutions of the BdG equation (14), $\Psi_\omega$ and $\Psi_{\omega'}$. By multiplying the equation $\Psi_{\omega'}$ on the left by $\Psi_\omega^\dagger$, and the equation for $\Psi_\omega^\dagger$ on the right by $\Psi_{\omega'}$, followed by their subtraction and integration with respect to $x$

results in

$$(\omega' - \omega) \int \mathrm{d}x \Psi_\omega^\dagger \sigma_3 \Psi_{\omega'} = 0, \tag{C1}$$

if the solutions are assumed to vanish at infinity. This is just the expression for the Bogoliubov scalar product in the space of classical solutions between any two such solutions. In particular, Eq. (C1) holds as long as

$$\int \mathrm{d}x \Big\{ \frac{1}{2} \partial_x [\Psi_\omega^\dagger \partial_x \Psi_{\omega'} - (\partial_x \Psi_\omega^\dagger) \Psi_{\omega'}] \\ + 3[\Psi_\omega^\dagger \sigma_4 \mathcal{G} * \Psi_{\omega'} - (\mathcal{G} * \Psi_\omega^\dagger) \sigma_4 \Psi_{\omega'}] \Big\} = 0. \tag{C2}$$

is fulfilled. As we show in the main text, the boundary conditions (27) for the approximated model mean exactly that we have $\mathcal{G} * \Psi_\omega = \tilde{\mathcal{G}}(-i\beta\partial_x)\Psi_\omega$ *at the system's solutions*. With this, it is straightforward to perform the integral in Eq. (C2):

$$\sum_{k,k'} S_k^* S_{k'} \Phi_k^\dagger \left[ \frac{k^* + k'}{2} - 3\sigma_4 \frac{\tilde{\mathcal{G}}(\beta k') - \tilde{\mathcal{G}}(\beta k^*)}{k' - k^*} \right] \Phi_{k'} = \sum_{p,p'} S_p^* S_{p'} \Phi_p^\dagger \left[ \frac{p^* + p'}{2} - 3\sigma_4 \frac{\tilde{\mathcal{G}}(\beta p') - \tilde{\mathcal{G}}(\beta p^*)}{p' - p^*} \right] \Phi_{p'}, \tag{C3}$$

where $k = k_\omega, k' = k_{\omega'}$, and similarly for $p, p'$. Equation (C3) includes the result, namely, Eq. (30) in the main text, which we are looking for. If we let $\omega' \to \omega$, the off-diagonal terms in Eq. (C3) alongside terms involving evanescent channels vanish in view of Eq. (15), and the remaining diagonal terms are precisely the flux conservation presented in Eq. (30) of the main text.

## Appendix D: S-matrix examples from solving Eq. (20)

In this appendix, selected explicit scattering coefficients of quasiparticle modes from solving the singular integral BdG equation (20) are worked out. As anticipated in the main text, solutions can be built numerically to any accuracy and precision, by calculating the various functions $\Lambda_{k,q}$ and $\Lambda_{k,q}$ and following the recipe developed in the main text Subsec. III C. In Fig. 10, we present some examples for solutions of (20), which are used to construct the Tables I, II, and III. The method we have implemented here is based on the spline technique of [49], which requires a higher computational effort, but on the other hand does not rely on prior assumptions on the form of the solutions besides continuity to hold for its effectiveness.

We present in Tables I, II, and III the S-matrix coeffi-

cients of three sets of quasiparticle modes in this system. All the matrices have been verified to satisfy the unitarity condition ($\boldsymbol{S}_\omega^\dagger \boldsymbol{S}_\omega = \mathbb{1}$) numerically. For the indicated parameters, the roton minimum exists and is given by $\Omega^{(\mathrm{r})} \sim 0.239701$, with $\Omega^{(\mathrm{m})} \sim 0.438915$. In particular for the quasiparticle modes in Table III, we set $\omega = 0.24$, which is close to the roton minimum frequency. These various coefficients give us information of how the barrier transmits and reflects for these three frequencies, as explained in the main text Subsec. V B.

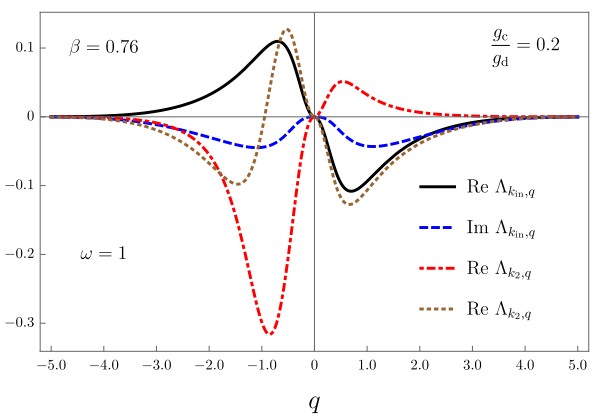

FIG. 10. Examples for solutions obtained from the integral equation (20), using the spline method of [49].

|  | $S_{k_1}$ | $S_{p_1}$ |
|---|---|---|
| $k_\text{in}$ | $0.070795 - 0.006271i$ | $-0.997384 - 0.013169i$ |
| $p_\text{in}$ | $-0.997384 - 0.013169i$ | $-0.070605 - 0.008138i$ |

TABLE I. S-matrix coefficients for a quasiparticle mode obtained from (20). Here we used as parameters $\omega = 1$, $g_\text{c}/g_\text{d} = 0.2$ and $\beta = 0.76$. For this frequency, there are no rotonic excitations involved, and thus the S-matrix is $2 \times 2$. The roton frequency is $\Omega^\text{(r)} \sim 0.239701$ and $\Omega^\text{(m)} \sim 0.438915$.

|  | $S_{k_1}$ | $S_{k_2}$ | $S_{k_3}$ | $S_{p_1}$ |
|---|---|---|---|---|
| $k_\text{in1}$ | $-0.38077 - 0.809387i$ | $0.33215 - 0.158392i$ | $-0.0072829 + 0.189553i$ | $-0.0899601 + 0.142897i$ |
| $k_\text{in2}$ | $-0.0751446 + 0.0993218i$ | $0.056732 + 0.0424557i$ | $0.33215 - 0.158392i$ | $-0.915897 - 0.0720336i$ |
| $k_\text{in3}$ | $-0.237571 - 0.181709i$ | $-0.0751446 + 0.0993218i$ | $-0.38077 - 0.809387i$ | $0.0257704 - 0.30704i$ |
| $p_\text{in}$ | $0.0257704 - 0.30704i$ | $-0.915897 - 0.0720336i$ | $-0.08996 + 0.142897i$ | $-0.159433 + 0.0841153i$ |

TABLE II. S-matrix coefficients for a quasiparticle mode obtained from (20). Here we used as parameters $\omega = 0.4$, $g_\text{c}/g_\text{d} = 0.2$ and $\beta = 0.76$. For this frequency, the rotonic excitations are involved, and thus the S-matrix is $4 \times 4$. The roton frequency is $\Omega^\text{(r)} \sim 0.239701$ and $\Omega^\text{(m)} \sim 0.438915$.

|  | $S_{k_1}$ | $S_{k_2}$ | $S_{k_3}$ | $S_{p_1}$ |
|---|---|---|---|---|
| $k_\text{in1}$ | $-0.996845 - 0.0690208i$ | $0.0108627 - 0.0284962i$ | $0.012463 + 0.00584869i$ | $-0.00187421 + 0.0203174i$ |
| $k_\text{in2}$ | $0.000119136 + 0.0218464i$ | $0.0597137 + 0.017794i$ | $0.0108626 - 0.0284962i$ | $-0.997201 - 0.017357i$ |
| $k_\text{in3}$ | $-0.0143499 + 0.00444707i$ | $0.00011914 + 0.0218464i$ | $-0.996845 - 0.0690208i$ | $-0.00791068 - 0.0277546i$ |
| $p_\text{in}$ | $-0.00791066 - 0.0277546i$ | $-0.997201 - 0.017357i$ | $-0.00187421 + 0.0203174i$ | $-0.0615123 + 0.0160217i$ |

TABLE III. S-matrix coefficients for a quasiparticle mode obtained from (20). Here we used as parameters $\omega = 0.24$, $g_\text{c}/g_\text{d} = 0.2$ and $\beta = 0.76$. For this frequency, the rotonic excitations are involved, and thus the S-matrix is $4 \times 4$. The roton frequency is $\Omega^\text{(r)} \sim 0.239701$ and $\Omega^\text{(m)} \sim 0.438915$.

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
