# Peer review of "Nonlocal field theory of quasiparticle scattering in dipolar Bose-Einstein condensates"

_SciPost Physics_

## Round 1 · Referee Report · Anonymous (Referee 1) · 2022-1-13

Strengths

The article attempts to address the problem of a domain wall of contact interaction in a one dimensional gas of dipolar atoms in the framework of the one-dimensional Gross Pitaevskii equation. It points out that the linearized Gross Pitaevskii equation is an integrodifferential equation with an integral kernel whose Fourier transform has a branch point for imaginary wavevector that gives rise to a continuum of evanescent waves near the discontinuity of contact interactions. It attempts to solve the linearized equation by singular integral equation techniques and discuss current conservation.

Weaknesses

1) The presentation of the manuscript is very sketchy. The potential energy terms are described in Eq. (4) for the dipolar interaction and the first paragraph of Sec. III-A for the discontinuous contact interaction. The full Gross-Pitaevskii equation (GPE) is never given, only the assumed linearized equation (8). However, that form is incorrect. The chemical potential $\mu$ is given by:
\begin{equation}
-\frac 1 2 \nabla^2 \psi_0 +(g_c(x)+g_d) |\psi_0|^2(x)\psi_0(x) -3 g_d \psi_0(x) \int dy G(x-y)|\psi_0|^2(y) = \mu \psi_0
\end{equation}
and substraction of the chemical potential from the linearized GPE yields
\begin{equation}
i \partial_t (\delta \psi) = -\frac 1 2 \nabla^2(\delta\psi) + \frac{\nabla^2 \psi_0}{2\psi_0} \delta\psi + (g_c(x) + g_d) |\psi_0(x)|^2 (\delta\psi+\delta \psi^*) \psi(x) -3 g_d \psi_0(x) \int dy G(x-y) \psi_0(y) (\delta\psi +\delta \psi^*) (y)
\end{equation}
One may use $\psi_0(y) \delta \psi$ as the new variable, but this would introduce extra terms coming from the $1/\psi_0(y)$ under the $\nabla^2$ differential operator.
To arrive at Eq. (8) onehas to neglect the full spatial variation of the stationnary solution $\psi_0(x)$, by assuming a step discontinuity in $x=0$. Even then, the integral kernal has to be split into two parts $y>0$ and $y<0$. The lack of translational invariance leads to equations more involved that Eq. (9), and the discussion in the rest of the paper lacks physical relevance.
2) A related issue is the discussion of the current conservation. The full GPE gives a conserved current that is exactly
\begin{equation}
j=\frac{\hbar}{2im}( \Psi^*\nabla \Psi -\Psi \nabla \Psi^*)
\end{equation}
and the linearized conserved current is obtained by inserting the expansion $\Psi=\psi_0+\delta \psi$. The conserved current derived from the linearized GPE should match that exact current. It is not clear that Eq. (25) satisfies that property.
3) Given that strong approximations have to be made to arrive at Eq. (8), it is not obvious that the sophisticated singular integral equation treatment of sec. III-C is physically relevant. Scattering processes taking place over a distance of the order of the healing length in the vicinity of the contact interaction discontinuity is neglected, yet they can contribute evanescent waves of larger range than the ones retained in Eq. (13).

Report

Given the weaknesses of the theoretical treatment of the Gross Pitaevskii equation, the physical relevance of the results in unclear. I cannot recommend publications in the present form. After a major revision, starting from a correct treatment of the static solution of the GPE followed by linearization to obtain a physically relevant integrodifferential equation, the manuscript could be reconsidered.

Requested changes

1) Formulate clearly the full Gross Pitaevskii equation with dipolar interaction and discontinuous contact interaction.
2) Derive the static solution
3) Consider the linearization around that static solution and discuss possible approximations in the obtained integrodifferential equation.
4) Discuss a simplified case with a step discontinuity in the static solution to check whether equations of the form (12) can be relevant or need already some modifications.

  • validity: low
  • significance: low
  • originality: ok
  • clarity: ok
  • formatting: good
  • grammar: good

Author:  Uwe R. Fischer  on 2022-01-19  [id 2109]

(in reply to Report 1 on 2022-01-13)

{\it Strengths: The article attempts to address the problem of a domain wall of contact interaction in a one dimensional gas of dipolar atoms in the framework of the one-dimensional Gross Pitaevskii equation. It points out that the linearized Gross Pitaevskii equation is an integrodifferential equation with an integral kernel whose Fourier transform has a branch point for imaginary wavevector that gives rise to a continuum of evanescent waves near the discontinuity of contact interactions. It attempts to solve the linearized equation by singular integral equation techniques and discuss current conservation.}

$\Longrightarrow$ We thank the Referee for reading our manuscript and pointing out its strengths in an organized manner. One of the major issues of the Referee's report, is that they suggest the manuscript did not present a clear definition of the conservation law being discussed. However, apparently the Referee refers to particle number conservation implied by the theory's $U(1)$ symmetry. The conservation law discussed in the present paper is {\em not} related to particle number conservation, as we address in detail below.
We also discuss separately each of the weaknesses claimed by the Referee.

{\it Weakness (1): The presentation of the manuscript is very sketchy. The potential energy terms are described in Eq. (4) for the dipolar interaction and the first paragraph of Sec. III-A for the discontinuous contact interaction. The full Gross-Pitaevskii equation (GPE) is never given, only the assumed linearized equation (8). However, that form is incorrect.}

$\Longrightarrow$ The main objective of our work is to study the solutions for the Bogoliubov-de Gennes (BdG) equation, Eq.~(8) of the submitted manuscript, and the Referee asserts that our assumed form for Eq.~(8) is not correct, which would raise serious concerns regarding the manuscript's physical validity. However, Eq.~(8) as we presented is indeed correct as we clarify now. We assumed throughout the work a stationary condensate of homogeneous particle density. The system inhomogeneity at the level of the linearized perturbation is, in our work, modeled by a variable contact interaction which changes the local sound velocity, for a fixed and constant condensate particle density. In this way, $\psi_{0}$ written by the Referee in their BdG equation is a constant, and the resulting equation is precisely the same as Eq.~(8) of the manuscript.

The Referee also pointed out that the manuscript does not present a deduction for Eq.~(8) from the GPE, which we believe prevented the Referee's understanding of how the condensate background is determined in the analysis of our model. We have amended the manuscript with the details of obtaining Eq.~(8), by showing how the external potential should be chosen in the experiment such that the condensate background is homogeneous. Furthermore, we also stress that, as mentioned in the manuscript, the main result of our work, namely, the recipe for how to construct solutions for the perturbations in a 1D condensate with dipolar interactions with a single sound interface, does not depend on the homogeneous condensate density assumption, and the same technique can be applied to a condensate with a variable particle density at $x=0$. We call attention of the Referee to the fact that the solutions we built explore precisely the splitting of integral kernel into two parts $x'<0$ and $x>0$, as shown in detail in the Appendices A and B of the submitted ms. For a condensate with a variable density modeling the sound barrier, exactly the same steps that lead to the singular integral equation (SIE) (15) can be used, with the difference that the resulting SIE depends explicitly on the density jump at $x=0$, and the potential term $\partial_x^2\psi_{0}/2\psi_{0}$ pointed out by the Referee becomes a delta derivative potential, explored, for instance, in [Ann. of Phys. {\bf 407,} 148-165 (2019)].

{\it Weakness (2): A related issue is the discussion of the current conservation. The full GPE gives a conserved current that is exactly and the linearized conserved current is obtained by inserting the expansion $\psi_0+\delta\psi$. The conserved current derived from the linearized GPE should match that exact current. It is not clear that Eq. (25) satisfies that property.}

$\Longrightarrow$ We thank the Referee for raising this point, whose detailed resolution we think contributes for our manuscript's clarity. The Bogoliubov expansion we use is known to lead to a spontaneous breakdown of the theory's $U(1)$ symmetry [cf. Phys. Rev. A {\bf 57}, 3008 (1998)], and thus there is no particle number conservation at the level of linearized $\delta\psi$ alone.
This can be deduced directly from the BdG equation, Eq.~(8) from the ms:

\begin{equation*}
i\partial_t\delta\psi=-\frac{\partial_x^2}{2}\delta\psi+n(g_{\rm c}+g_{\rm d})(\delta\psi+\delta\psi^*)-3ng_{\rm d}[G*(\delta\psi+\delta\psi^*)].
\end{equation*}

Multiplying the equation above by $\delta\psi^*$, and subtracting the result from its complex conjugate results in

\begin{equation*}
\partial_t|\delta\psi|^2+\frac{1}{2i}\partial_x[\delta\psi^*\partial_x\delta\psi-(\partial_x\delta\psi^*)\delta\psi]=\frac{n(g_{\rm c}+g_{\rm d})}{i}(\delta\psi^2-\delta\psi^{*2})-3ng_{\rm d}\frac{\delta\psi^*-\delta\psi}{i}[G*(\delta\psi+\delta\psi^*)].
\end{equation*}

The L.H.S. of the equation above contains the contribution to the full particle density coming from the perturbations: $\delta n=|\delta\psi|^2$, and the corresponding contribution to the system full particle current: $\delta J=\mbox{Im}\ \delta\psi^*\partial_x\delta\psi$. Yet, even in the absence of dipolar interactions, $g_{\rm d}=0$, we find $\partial_t\delta n+\partial_x\delta J\neq0$, which is the mathematical expression for the no number conservation of the expansion. A thorough study of the physical meaning of this symmetry breakdown by the Bogoliubov expansion and how to correctly address the corrections to the condensate due to the depleted particles can be found, for instance, in Phys. Rev. D {\bf 72}, 105005 (2005).

However, and this seems to be the root of the Referee's stated problem, there exists a conserved quantity implied by Eq.~(8) which is linked to the exact {\em unitarity of the scattering process,} and not to the theory's number conservation. Specifically, the BdG equation in Nambu representation, $\Psi=(\delta\psi, \delta\psi^*)^{\rm t}$, reads

\begin{equation*}
i\sigma_3\partial_t\Psi=-\frac{\partial_x^2}{2}\delta\Psi+n(g_{\rm c}+g_{\rm d})(1+\sigma_1)\Psi-3ng_{\rm d}(1+\sigma_1)G*\Psi,
\end{equation*}

which is the equation solved in the ms. In order to deduce the conserved ``charge'' implied by the equation above, we let $\Psi$ be any solution, not necessarily satisfying $\sigma_1\Psi^*=\Psi$. By repeating the same steps as before, we multiply on the left by $\Psi^{\dagger}$ and subtract the complex conjugate to obtain

\begin{equation*}
\partial_t(\Psi^{\dagger}\sigma_3\Psi)+\frac{1}{2i}\partial_x[\Psi^{\dagger}\partial_x\Psi-(\partial_x\Psi^\dagger)\Psi]= 3ing_{\rm d}[\Psi^\dagger(1+\sigma_1)G*\Psi-(G*\Psi^\dagger)(1+\sigma_1)\Psi].
\end{equation*}

We observe that in the absence of dipolar interactions, a local conservation law exists: $\partial_t(\Psi^\dagger\sigma_3\Psi)+\partial_x\mbox{Im}\ \Psi^{\dagger}\partial_x\Psi=0$. This is the conservation law discussed in section IV of the ms, which assumes the form $\partial_xJ_{\omega}=0$ for the quasiparticle excitations in the absence of nonlocal interactions. For the dipolar system in the ms, however, the local conservation law no longer holds, but the full ``charge'' $\int dx \Psi^{\dagger}\sigma_3\Psi$ is stil conserved in time, as can be seen from the integration of the equation above. Furthermore, by following the same steps that lead to the equation above for any two solutions $\Psi$, $\Psi'$, we show that the BdG scalar product $$\langle\Psi,\Psi'\rangle:=\int dx\Psi^\dagger\sigma_3\Psi'$$ is conserved in time: $\partial_t\langle\Psi,\Psi'\rangle=0$. For the quasiparticle excitations, this condition is equivalent to the ``conservation law'' stated in the manuscript, from which we can assess the unitarity of the scattering process, i.e., the
balance between incident, reflected, and transmitted waves.
We cite in this regard two references
[Phys. Rev. D {\bf 93}, 124060 (2016), Phys. Rev. D {\bf 94}, 084027 (2016)] for a discussion of the scattering process in the context of analogue gravity.
By verifying the S-matrix unitarity for the built quasiparticles, we thus ensure that the field modes we constructed are indeed solutions to the BdG equation. We have amended the manuscript by stating explicitly the meaning of the S-matrix in terms of the Bogoliubov scalar product time-invariance.

{\it Weakness (3): Given that strong approximations have to be made to arrive at Eq. (8), it is not obvious that the sophisticated singular integral equation treatment of sec. III-C is physically relevant. Scattering processes taking place over a distance of the order of the healing length in the vicinity of the contact interaction discontinuity is neglected, yet they can contribute evanescent waves of larger range than the ones retained in Eq. (13).}

$\Longrightarrow$ As explained in the above, our development relies on no approximation besides the Bogoliubov expansion to linearize the GPE and study its perturbations, and thus Eq.~(8) is exact for the condensate with constant particle density we assumed. The implementation of our technique to the variable density case is a straightforward extension that leads to no qualitative differences to the scattering phenomena we have discovered, and whose addition to the manuscript would only further increase the already rather elevated mathematical complexity of the problem.

---

## Round 2 · Referee Report · Anonymous (Referee 3) · 2022-2-23

Strengths

1) The authors propose a general method to consider the effect of long range interactions in linearized Gross-Pitaevskii equations. They are able to show that the solution takes the form of plane waves plus evanescent waves and derive the scattering coefficients in the presence of a roton minimum in the dispersion at the left of the defect.
2) They have also introduced a numerical approach relying on the discretization of the Fourier transform of the long range interaction potential, turning the branch cut into a sum over discrete poles.

Weaknesses

1) The authors are choosing the external potential in Eq. (10) in such manner that the Gross-Pitaevskii equation (9) has a uniform solution for a particular density. This allows plane/evanescent wave solutions to the linearized equation (12) discussed in the rest of the manuscript, but the treatment applies only at the special density such that $U(x)+ng_c(x)$ is a constant. In the case of a more general potential, the average density $n(x)$ will vary around the defect over a length of order the healing length. This is likely to hamper the construction of the asymptotic states starting from Eq. (11).
2) It appears that in Eq. (B2), the integral over q' has to be taken in principal value due to the presence of the pole at q'=q. The contribution of the pole is $\bar{G}$ in Eqs. (20) and (B2). Eq. (B8) is also a relation between principal values. This should be stated explicitly since the extra delta functions contribution that would arise from regularizing asymetrically Eq. (B8) by adding a positive infinitesimal imaginary part to $q_\pm$ would spoil the boundary conditions (21) and (22).

Report

The authors have clarified their choice of external potential and the meaning of the current conservation in Eq. (27). While the solution is limited to a particular choice of potential to ensure a constant density in the time independent solution, the solution for the linearized mode is valid and should be applicable to other models of integrodifferential equations with translationally invariant Kernel and inhomogeneous boundary conditions.
I would thus incline to recommend publication of the manuscript.

  • validity: good
  • significance: ok
  • originality: good
  • clarity: good
  • formatting: excellent
  • grammar: good

Author:  Uwe R. Fischer  on 2022-02-26  [id 2247]

(in reply to Report 1 on 2022-02-23)

We thank the Referee for the careful reading of the manuscript, and for providing an insightful report.
We respond to the two weakness points as follows.

Weakness 1: We thank the Referee for raising this issue, and we agree: The solutions we constructed are accurate for the particular constant density profile we analyzed, and sound barriers produced by continuous variations of the system density are likely to produce quantitatively different results. We believe the generalization of our technique to condensates with continuously varying parameters density and coupling deserves a dedicated analysis, and we amended the manuscript conclusion by mentioning the corresponding further perspective.

Weakness 2: We thank the Referee for noting this aspect in our manuscript. Indeed, the Cauchy principal value is to be taken for the integral wrt $q'$ in Eq. (B2), and this was already noted before in the main text after Eq. (19). We amended the manuscript to state that also explicitly after Eq. (B2).

---

## Round 2 · Referee Report · Anonymous (Referee 4) · 2022-6-21

Strengths

1-First paper I am aware of to consider effect of roton in dispersion relation to problems involving inhomogeneous speed of sound.
2- Nice discussion of conserved currents and related issues for BdG problem given in response to another referee comment. It might be useful to include that in the paper .

Weaknesses

1-I found this paper hard to read and there was not much intuition/explanation provided of the physics involved. What is the significance of the evanescent modes? Why do they particularly appear in the presence of long-range/anisotropic interactions?
2-I think the non-analyticity of the kernel could be better explained. I note that equation (3) shows that the dipolar interaction in momentum space is better behaved than in real space since it is purely angular: 3Cos^2 (theta) -1. This seems easier to deal with than the 1/r^3 divergence in coordinate space. Please explain why you go into the complex plane to search out the poles.
3-What is the significance of the ansatz (16) and the accompanying equations (17) and (18)? Is this standard or is it something radical?
4-Eq (19) is also mysterious to me and it would be useful for it to be explained. How unusual is this equation? (In comparison to contact case, say.) What, physically, are the solutions Lambda in this equation? Are they propagators?
5- Section III D on approximate solutions confuses me, especially the 2N solutions. What is Fig 4 telling us?
6- Discussion of the physical implications of the results shown in Figs 5,6,7,8,9 would be useful.
7- the term hypersingular is mentioned in the abstract and the introduction but I am not sure what the significance of this is for the current problem and how special or unusual it is.

Report

Speaking as someone who has worked both on dipolar BECs and dabbled in analogue Hawking radiation, I still found this paper hard to understand even though it is ``in my area''. The work is a generalization to the case of dipolar interactions of work on sound barriers in standard BECs with contact interaction and is related to analogue Hawking radiation experiments. The authors make some interesting points since the roton minimum in the excitation spectrum allows for waves with strange properties and is probably correct, but for my taste not enough physical intuition is given on the results (and hence I find them hard to appreciate). I did not find the presentation very clear and for that reason I would not recommend publication in SciPost Physics. I think it meets the criteria for publishing as a standard article (in the Core journal) . I think the basic premise is interesting and there may well be other groups who follow up on this work

Requested changes

See section on "weaknesses" for suggested changes.

---

## Round 2 · Author Response

We have responded to the existing Referee report in detail previously.

---

## Round 2 · List of Changes

The changes we performed are already detailed in the response letter to the Referee.
In summary, they are

On the Referee point 1: With reference to the claim of the Referee that we performed an approximation by neglecting density gradients, which is not the case.
Answer: We reworked section IIIA, in which we now explicitly demonstrate the
order parameter stationary solution for our homogeneous density condensate, by amending the manuscript with the details of obtaining Eq. (8), showing how the external potential should be chosen in the experiment such that the condensate background is homogeneous.

On the Referee point 2: The alleged problem related to particle number conservation.
We addressed this by adding a paragraph added after Eq. (27), explaining that the conservation law we have is not due to the U(1)-symmetry related particle conservation of the full theory.

On the Referee point 3: This is in reality point 1 reiterated, so there is no need of an additional change. Our development relies on no approximation besides the Bogoliubov expansion to linearize the Gross-Pitaevskii equation and study its perturbations, and thus Eq. (8) is exact for the condensate with constant particle density we assumed.

---

## Editorial Decision

resubmitted